# An extended DNA-free intranuclear compartment organizes centrosome microtubules in malaria parasites

Caroline S Simon[1] , Charlotta Funaya[2] , Johanna Bauer[1] , Yannik Voβ[1], Marta Machado[1,3], Alexander Penning[1] , Darius Klaschka[1], Marek Cyrklaff[1], Juyeop Kim[1] , Markus Ganter[1] , Julien Guizetti[1]

Proliferation of *Plasmodium falciparum* in red blood cells is the cause of malaria and is underpinned by an unconventional cell division mode, called schizogony. Contrary to model organisms, *P. falciparum* replicates by multiple rounds of nuclear divisions that are not interrupted by cytokinesis. Organization and dynamics of critical nuclear division factors remain poorly understood. Centriolar plaques, the centrosomes of *P. falciparum*, serve as microtubule organizing centers and have an acentriolar, amorphous structure. The small size of parasite nuclei has precluded detailed analysis of intranuclear microtubule organization by classical fluorescence microscopy. We apply recently developed super-resolution and time-lapse imaging protocols to describe microtubule reconfiguration during schizogony. Analysis of centrin, nuclear pore, and microtubule positioning reveals two distinct compartments of the centriolar plaque. Whereas centrin is extranuclear, we confirm by correlative light and electron tomography that microtubules are nucleated in a previously unknown and extended intranuclear compartment, which is devoid of chromatin but protein-dense. This study generates a working model for an unconventional centrosome and enables a better understanding about the diversity of eukaryotic cell division.

## Introduction

*Plasmodium falciparum* encounters significant population bottlenecks when being transmitted between humans and mosquitoes. To overcome those, it undergoes several phases of extensive proliferation. When a mosquito takes up an infected blood meal, a rapid series of division events is triggered resulting in the formation of eight male gametes from a single gametocyte within only 15 min (Sinden et al, 1978; Fang et al, 2017). After fusion of male and female gametes, the resulting ookinete penetrates the mosquito midgut to form an oocyst. During this stage, thousands of sporozoites are produced from a single progenitor cell (Beier, 1998; Vaughan, 2007).

After sporozoites reach the salivary gland of the mosquito, they can be injected into humans during a bite. Once those sporozoites invade a hepatocyte, they can generate more than 10,000 daughter cells within one cycle, which are then released into the blood (Prudêncio et al, 2006; Sturm et al, 2006). There, repeated rounds of red blood cell invasion, growth, division, and egress cause the high parasite loads, which lead to all clinical symptoms associated with malaria (Schofield, 2007) (Fig 1A). The cell division processes that underlie these unconventional proliferation events are, however, poorly understood (Francia & Striepen, 2014; Matthews et al, 2018; Gubbels et al, 2020; Simon et al, 2021).

Successful division requires a series of cellular events. Chromosomes must be replicated alongside duplication of the centrosomes, which act as the poles towards which sister chromatids are segregated. Thereafter nuclei are physically separated and the cytoplasm is divided by cytokinesis. *P. falciparum*, however, uses an unconventional division mode, called schizogony, where several rounds of nuclear divisions are not interrupted by cytokinesis (Fig 1A), leading to formation of multinucleated parasite stages (Leete & Rubin, 1996). Although nuclei share a common cytoplasm, nuclear divisions are asynchronous (Read et al, 1993; Arnot et al, 2011; Dorin-Semblat et al, 2011). Throughout nuclear division, the nuclear envelope remains intact and the DNA is not condensed (Read et al, 1993). Once all rounds of nuclear division are completed, each of the 8–28 nuclei are packaged into individual daughter cells, called merozoites (Reilly et al, 2007; Garg et al, 2015; Rudlaff et al, 2019; Simon et al, 2021). Upon rupture of the infected host cell, merozoites are released and invade new red blood cells (Fig 1A).

Centrosomes are generally regarded as key regulatory hubs of the cell cycle, and their duplication limits the number of nuclear divisions (Fu et al, 2015). The centrosome of *P. falciparum* is called the centriolar plaque (Sinden, 1991; Arnot et al, 2011). It exhibits important morphological differences when compared with model organisms such as vertebrate centrosomes or the spindle pole bodies in yeast (Rüthnick & Schiebel, 2018). Available data on the organization of the centriolar plaque are very limited. In early transmission EM studies, mainly carried out in oocysts in the mosquito midgut, centriolar plaques appear as electron-dense

---

[1]Centre for Infectious Diseases, Heidelberg University Hospital, Heidelberg, Germany  [2]Electron Microscopy Core Facility, Heidelberg University, Heidelberg, Germany [3]Graduate Program in Areas of Basic and Applied Biology, Instituto de Ciências Biomédicas Abel Salazar, Universidade do Porto, Porto, Portugal

Correspondence: julien.guizetti@med.uni-heidelberg.de

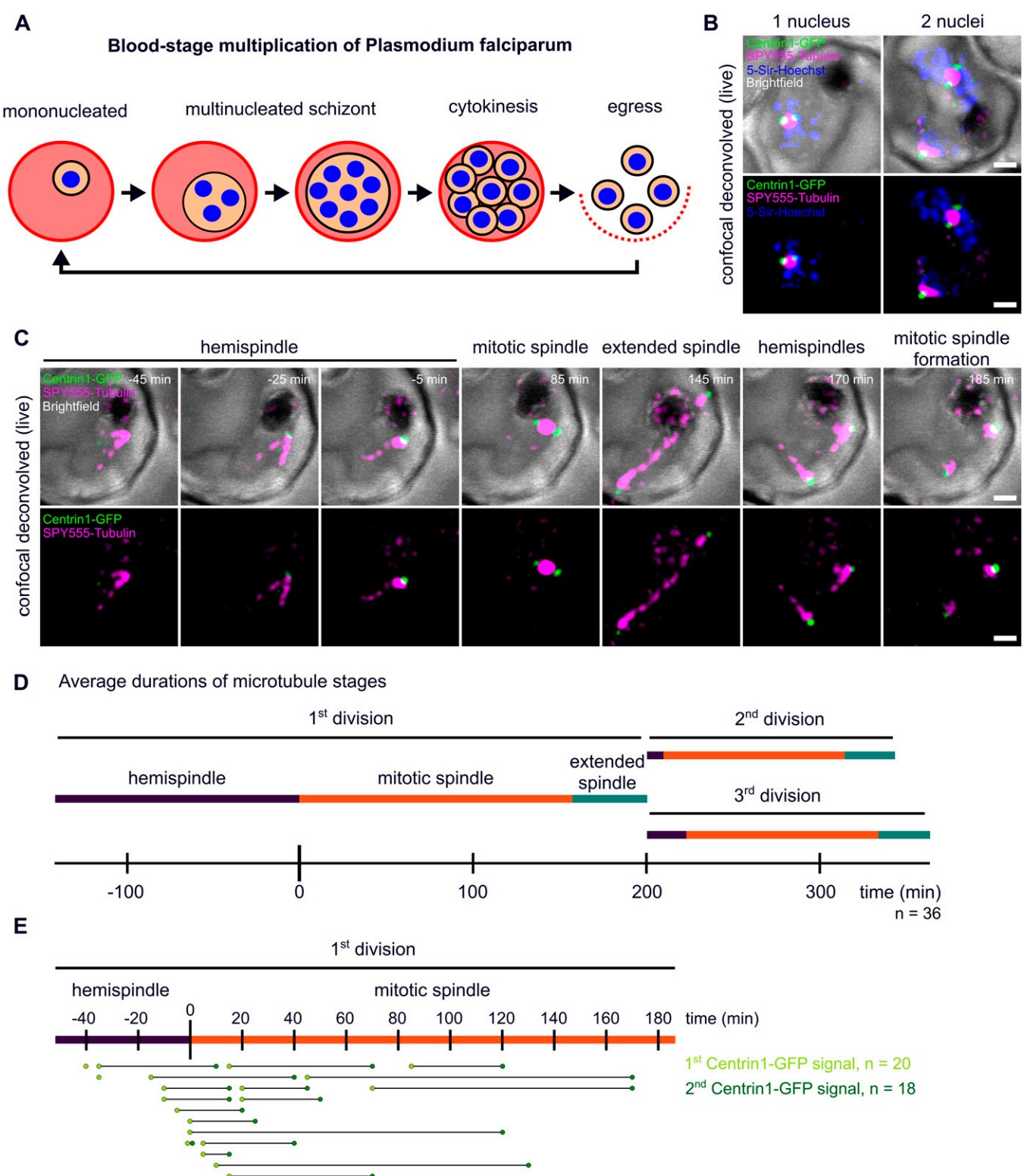

**Figure 1. Live-cell imaging of microtubule and centriolar plaque reorganization throughout schizogony.**
**(A)** Schematic of *P. falciparum* blood-stage development including multiple divisions (schizogony) before cytokinesis and egress of new infectious parasites.
**(B)** Deconvolved confocal live-cell still images of two separate *Plasmodium* NF54 infected red blood cells ectopically expressing PfCentrin1-GFP (green), labeled with SPY555-Tubulin (magenta) and 5-Sir-Hoechst (blue). The images are maximum intensity projections. **(C)** Time-lapse of a cell labeled as in (B), but without 5-Sir-Hoechst. The first spindle formation and elongation in a single parasite is shown over time. **(D)** Quantification of average duration of three distinct microtubule organization stages in 36 cells (acquired in three replicates). Because most movies (n = 32/36) were already started at hemispindle stages, we could only quantify the minimal mean length of the hemispindle stage for the first division. **(E)** Time points of appearance of first (n = 20, three replicates) and second (n = 18, three replicates) clear PfCentrin1-GFP signals normalized to the start of accumulating tubulin signal (start of mitotic spindle formation). All scale bars are 1 μm.

areas that neither show centrioles nor any other distinct structures (Aikawa et al, 1967; Terzakis et al, 1967; Aikawa & Beaudoin, 1968; Howells & Davies, 1971; Canning & Sinden, 1973; Sinden et al, 1976; Schrevel et al, 1977). Centriolar plaques seem partially embedded in the nuclear membrane, but their positioning relative to the nuclear pore-like "fenestra" remains unclear (Aikawa & Beaudoin, 1968; Wall et al, 2018; Zeeshan et al, 2020a). Generally, the amorphous appearance of centriolar plaques in EM has precluded a detailed analysis of their organization so far. The centrosome of a related apicomplexan parasite *Toxoplasma gondii*, which does contain centrioles, shows a bipartite organization with a distinct inner and outer core (Suvorova et al, 2015). Furthermore, few canonical centrosome components are conserved in *Plasmodium* with the exception of centrins, a family of small calcium-binding proteins implicated in centrosome duplication (Azimzadeh & Bornens, 2007; Mahajan et al, 2008; Roques et al, 2019), and the microtubule nucleating complex around γ-tubulin (Zupa et al, 2020). Centrins have been used as bona fide centrosome marker and shown to localize distinctly for the nuclear DNA, possibly embedded in the nuclear envelope (Mahajan et al, 2008). γ-tubulin has been shown to decorate the minus ends of subpellicular microtubules during blood-stage parasite cytokinesis (Fowler et al, 2001), but its localization with respect to intranuclear microtubules has not been analyzed. More canonical centriole components such as SAS-6 are also coded in the genome and likely contribute to the formation of basal bodies in microgametes, whereas being non-essential in the blood stage (Francia et al, 2015; Marques et al, 2015; Suvorova et al, 2015). Before we begin to understand the regulation of centriolar plaque duplication and nuclear division, we must know the arrangement and dynamics of key division factors around this atypical centrosome.

Centriolar plaques act as microtubule organizing centers. During schizogony, intranuclear microtubule organization is very heterogeneous and several atypical structures such as plaques, hemispindles, and the particularly small mitotic spindles have been described (Read et al, 1993; Fennell et al, 2006, 2008; Arnot et al, 2011). Tubulin-rich plaques were equated with the centriolar plaques, but their size clearly exceeds the dimensions of electron-dense regions described in EM studies (Gerald et al, 2011). Hemispindles were often interpreted as half-spindles that form a bipolar mitotic spindle by fusion (Schrevel et al, 1977; Read et al, 1993; Fennell et al, 2006, 2008). Inconsistencies in the observed size bring fusion of hemispindles during schizogony into question. While hemispindles observed in EM are about 0.5–0.7 $\mu$m in length, which is more consistent with an early stage of a mitotic spindle, hemispindles described by tubulin antibody staining are extensive structures measuring around 2–4 $\mu$m (Gerald et al, 2011). In another study, hemispindles observed in oocysts were interpreted as postanaphase spindles (Canning & Sinden, 1973). More recent fluorescence live-cell imaging data of the microtubule-associated protein Kinesin-5 in asexual and sexual stages demonstrated the dynamic elongation of the mitotic spindle, while hemispindles were not resolved (Zeeshan et al, 2020a). These data provide a controversial view on the occurrence, dynamics, and function of hemispindles, which needs to be clarified. We have recently established Stimulated Emission Depletion (STED) nanoscopy

for blood stages, which allowed us to resolve distinct microtubule nucleation sites (Mehnert et al, 2019). These findings were recently confirmed by ultrastructure expansion microscopy (Gambarotto et al, 2019; Bertiaux et al, 2021; Guizetti & Frischknecht, 2021). Where microtubule nucleation sites are positioned relative to the nuclear envelope is still an open question.

In this study, we use a combination of super-resolution, live-cell, and electron microscopy to reveal the ultrastructural organization of the centriolar plaque and microtubules in dividing *P. falciparum* blood-stage parasites. We characterize their unconventional dynamics and reveal a novel protein-dense subnuclear compartment that harbors microtubule nucleation sites and is devoid of chromatin.

## Results

To analyze centriolar plaque and microtubule dynamics, we carried out live-cell imaging of nuclear divisions in a blood-stage *P. falciparum* parasite strain that episomally expresses PfCentrin1-GFP and was labeled with SPY555-Tubulin, a live-cell–compatible microtubule dye (Wang et al, 2020), alongside 5-SiR-Hoechst, a fluorogenic infrared DNA dye (Bucevičius et al, 2019) (Fig 1B). Consistent with the specificity of the dyes, every schizont nucleus displayed a microtubule staining. However, even small concentrations of any live-cell DNA dye we tested can inhibit mitotic progression. Therefore, we omitted the DNA dye for time-lapse microscopy (Fig 1C). To reduce phototoxicity, we applied gentle illumination conditions and used HyVolution-based image processing to generate sufficient image contrast to detect those weak signals (Video 1). We selected individual cells with a single SPY555-Tubulin signal (Fig 1C), indicating the presence of a single nucleus at the start of schizogony (Fig 1B). Initially, microtubules dynamically extended forming hemispindle structures (Fig 1C). This was followed by a prolonged phase, where the tubulin signal accumulated close to the PfCentrin1-GFP signal (time = 0 min) to form a mitotic spindle after centrin duplication. Finally, the tubulin signal elongated, whereas the distance between centrin foci increased leading to an extended spindle. We quantified the mean duration of hemispindle, mitotic spindle, and extended spindle stages throughout the first three nuclear divisions (Fig 1D). Every microtubule organization stage was significantly longer in the first division when compared with the second or third (Fig S1A). The precise time point of initial appearance of the centrin focus varied and was sometimes difficult to determine depending on PfCentrin1-GFP expression levels (Fig 1E). In cells with a stronger signal, appearance of the centrin focus already coincided with the hemispindle microtubule stage (Fig 1C). In other cells it was only detectable later (Video 2) or rarely not at all. A second centrin focus appeared about 65 min on average after the first one (Fig S1B). This was promptly followed by elongation of microtubules indicating that a mitotic spindle was assembled during centrin signal duplication. The now segregated centrin foci were again associated with dynamic hemispindle structures, which subsequently went through a collapsed stage before initiating the next duplication and elongation event. Frequently, the second and third round of spindle elongation occurred in an asynchronous fashion (Fig 1D).

Because of the inherent resolution and sensitivity limits of live-cell imaging, we could not consistently determine the appearance of the centrin signal and its duplication. Furthermore, we wanted to analyze organization of endogenous centrin and its exact positioning relative to microtubules. Therefore, we imaged parasites at different stages of schizogony after immunolabeling with an anti-tubulin antibody and a newly generated anti-centrin antibody (Fig S2) by dual-color STED nanoscopy (Fig 2A). Although early trophozoite-stage parasites do not express tubulin or centrin (Fig S3), we identified hemispindle structures already in mononucleated late trophozoites (Fig 2A). Consistent with the late appearance of PfCentrin1-GFP in our live-cell imaging data, only 24 out of 52 analyzed hemispindles in mononucleated cells were associated with an endogenous centrin signal, whereas in later stages, after the first division, every nucleus was accompanied by one or two

centrin foci. Higher sensitivity of immunofluorescence labeling shows that endogenous centrin accumulates before formation of the first mitotic spindle. The collapse of hemispindles into a more intense and compact mitotic spindle is associated with two centrin foci (Fig 2A). Because individual microtubules cannot be resolved within this structure, we cannot definitely determine when they are reorganized into the bipolar microtubule array, which makes up the mitotic spindle. However, early stages where duplicated centrin foci are proximal can be differentiated from late stages where they oppose each other with DNA in the middle. Occasionally, two nuclei connected by an extended spindle can be observed (Fig 2A). In multinucleated stages, several different microtubule organizations could be observed simultaneously, reflecting the asynchrony of nuclear divisions. At no point did we detect astral or extranuclear microtubules in schizonts.

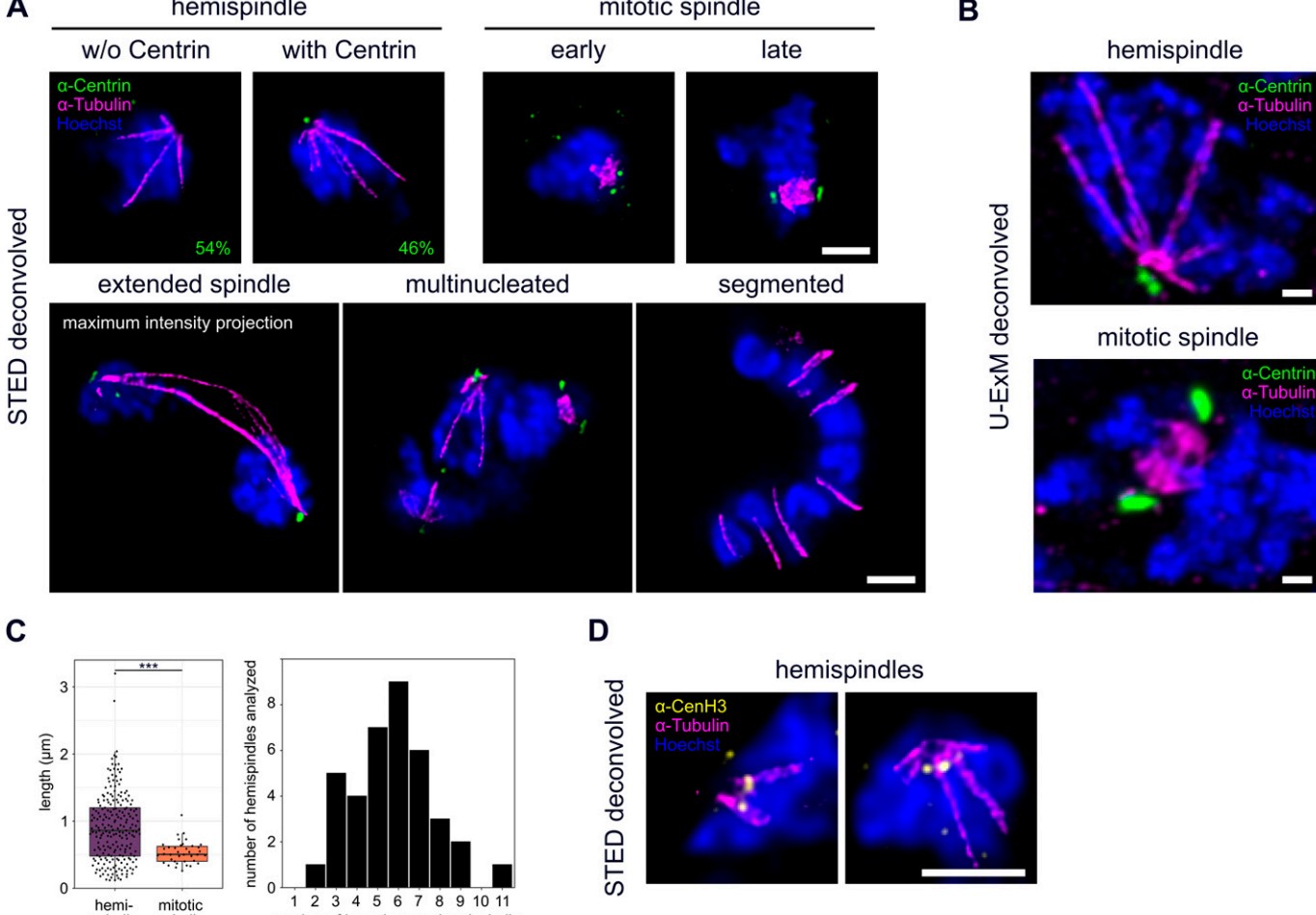

**Figure 2. STED super-resolution and ultrastructure expansion microscopy reveal detailed organization of microtubules, centriolar plaques and centromeres during schizogony.**

**(A)** Dual-color STED nanoscopy images of different schizogony stages of 3D7 parasites expressing tagged nuclear pore protein Nup313-HA_glms, labeled with anti-centrin (green), anti-tubulin (magenta) antibodies and stained with Hoechst (blue). Single slices are shown except for the extended spindle. Quantification of percentage of hemispindles in mononucleated cells with and without centrin signal was performed in 3D7 wild-type cells (n = 52 cells, 1 replicate) imaged with confocal microscopy. **(B)** Confocal U-ExM images of individual schizont nuclei of the 3D7 Nup313-3xHA_glms strain in hemispindle and mitotic spindle phase, labeled as in (A), except for using three instead of one anti-tubulin antibody. Maximum intensity projections are shown. **(C)** Quantification of lengths (n = 217, corrected by a measured expansion factor of 4.5) and number of hemispindle branches per nucleus (n = 38) and mitotic spindle lengths (n = 38) of 3D7 Nup313-3xHA_glms expressing cells imaged with U-ExM in 2 replicates. **(D)** Like (A) with anti-tubulin (magenta) and anti-CenH3 (yellow) showing centromere positioning in hemispindle phases. All scale bars are 1 $\mu$m.

Cytokinetic segmented stages on the other hand lack distinct centrin foci and intranuclear microtubules, but clearly display the microtubule cytoskeleton associated with the inner membrane complex (Fig 2A), a double-membrane organelle forming beneath the plasma membrane during the budding phase (Ferreira et al, 2021).

STED microscopy is, however, partly due to high laser intensities, significantly limited in the acquisition of full z-stacks of entire cells. To reveal detailed three-dimensional organization of spindles in dividing nuclei, we used ultrastructure expansion microscopy (U-ExM), which causes an isotropic expansion of immunolabeled cells (Gambarotto et al, 2019). 3D-rendering of acquired image stacks showed the radial branching of hemispindles (Video 3) and the compact organization of mitotic spindles (Video 4). Slices of those images reveal the same nuclear, microtubule, and centrin organization details as the STED images (Fig 2B). U-ExM data allowed reliable 3D length measurements of mitotic spindles averaging at about 560 nm (Fig 2C). For individual nuclei with hemispindles, the length of branches varied substantially with some clearly exceeding the nuclear diameter, whereas the number of branches per nucleus ranged from 2 to 11 (Fig 2C). In a classical conformation, we would expect microtubule minus end to be facing the centrosome. To test this, we attempted multiple labeling strategies for γ-tubulin, which decorates microtubule minus ends and promotes their nucleation. However, we only found one γ-tubulin antibody with acceptable background levels (Fig S4). Despite additional unspecific binding to microtubules, we found accumulations of γ-tubulin at the poles of the spindles.

Because the role of hemispindles is unclear, we wanted to test whether they might be involved in recruitment of centromeres, akin to a "search-and-capture" mechanism (Heald & Khodjakov, 2015), which could assist their clustering at the nuclear periphery (Hoeijmakers et al, 2012). Therefore, we co-labeled tubulin with an anti-CenH3 antibody, which specifically marks centromeric histones (Fig 2D). As expected, the centromere signal clustered at the periphery next to the centriolar plaque (Zeeshan et al, 2020b). However, the increased resolution of STED nanoscopy reveals individual CenH3 foci that were virtually always distinct from hemispindle microtubules, precluding any direct interaction at the acquired time points.

Notably, all images revealed a significant gap between centrin and tubulin signals (Fig 2A), which has been described previously (Roques et al, 2019; Bertiaux et al, 2021). The question whether microtubules and centrin locate inside or outside the nucleus remains open. In absence of a known nuclear envelope marker for *Plasmodium* spp., we used a strain where the nuclear pore protein Nup313, a likely component of the central FG-Nups layer, has been tagged endogenously with 3xHA (Fig S5) to partly mark the nuclear boundary (Kehrer et al, 2018). Immunofluorescence co-staining with centrin and tubulin revealed that centrin signals localize on the cytoplasmic side, whereas microtubule ends are localized inside the nucleus during all stages of schizogony (Fig 3A). We, also, consistently observed a Hoechst-free region right beneath the centrin foci, which has not been described previously. In nuclei with accumulated tubulin signals those localize within this region.

To test if this region is indeed devoid of DNA and not a result of inefficient Hoechst labeling, we stained cells with DRAQ5, which is an intercalating DNA dye not sensitive to heterochromatin state (Fig

3B). This staining confirmed the absence of DNA from this region beneath centrin. To further assess the position of the nuclear boundary at the centriolar plaque, we used a strain ectopically expressing mCherry tagged with three NLS to stain the nucleoplasm (Fig 3B) (Klaus et al, 2021 *Preprint*). The NLS-mCherry signal was overlapping with the DNA-free region suggesting that there is an extended subnuclear compartment devoid of DNA associated with the centriolar plaque. Measurements of this region indicate that its dimensions are not significantly different in hemispindle and mitotic stage nuclei (Fig 3C).

To more directly visualize the nuclear membrane and the ultrastructural features surrounding the centriolar plaque, we used EM. Initial analysis of schizont nuclei with transmission EM suggested that an intranuclear region associated with microtubules might, indeed, be delineated by a non-membranous boundary (Fig 3D). However, because of their amorphous structure, centriolar plaques cannot be consistently identified in those EM samples. Hence, we adapted an in-resin correlative light and electron microscopy (CLEM) approach (Kukulski et al, 2011, 2012) to our system using the PfCentrin1-GFP–expressing parasite line. In addition, we labeled the cells with the infrared DNA dye 5-SiR-Hoechst (Bucevičius et al, 2019). The fluorescent signal was preserved in the samples prepared for EM and resin sections were imaged on a widefield fluorescence microscope to identify the residual PfCentrin1-GFP foci (Fig 3E). Using overview images and finder grids, we were able to relocate individual cells at the electron microscope. Overlaying the fluorescence image with the electron tomogram allowed us to unambiguously define the centriolar plaque position. 5-SiR-Hoechst signal was also still detectable after sample preparation but, likely due to imaging a limited section of the nucleus, the staining was not uniform and mostly limited to electron-dense heterochromatin regions. The region associated with the centriolar plaque was consistently free from Hoechst staining (Fig 3E). We could never detect any invagination of the nuclear membrane adjacent to the centrin signal. However, we identified an underlying region with distinct electron density distribution, which was sometimes associated with microtubules (Fig 3E). The size and shape of that region corresponded well to the Hoechst-free region measured in our immunofluorescence staining (Fig 3C).

To reveal additional morphological features of the centriolar plaque, we applied an U-ExM protocol in which we labeled proteins in bulk using an NHS-ester dye conjugate together with tubulin (Fig 3F) and centrin (Fig 3G) antibody staining (Bertiaux et al, 2021). This revealed an "hourglass-shaped" highly protein-dense structure at the centriolar plaque region. Because the outer part of this structure colocalizes with centrin and the inner part with tubulin, we can assume that it stretches from the cytoplasm to the nucleoplasm. The most protein-dense part of this structure is formed by the "neck" which sits between the outer and inner parts. Mitotic spindles themselves also displayed an increased protein density. Whereas the outer part of the "hourglass-shape" was more irregularly shaped, we could measure the dimensions of the inner region, particularly when associated with hemispindles (Fig 3H). Whereas the NHS-ester signal that filled the intranuclear region had the same dimensions as the DNA-free region (Fig 3C), the dimensions of the highly protein-dense part was smaller (Fig 3H). Taken together, these data suggest that there is, indeed, a novel

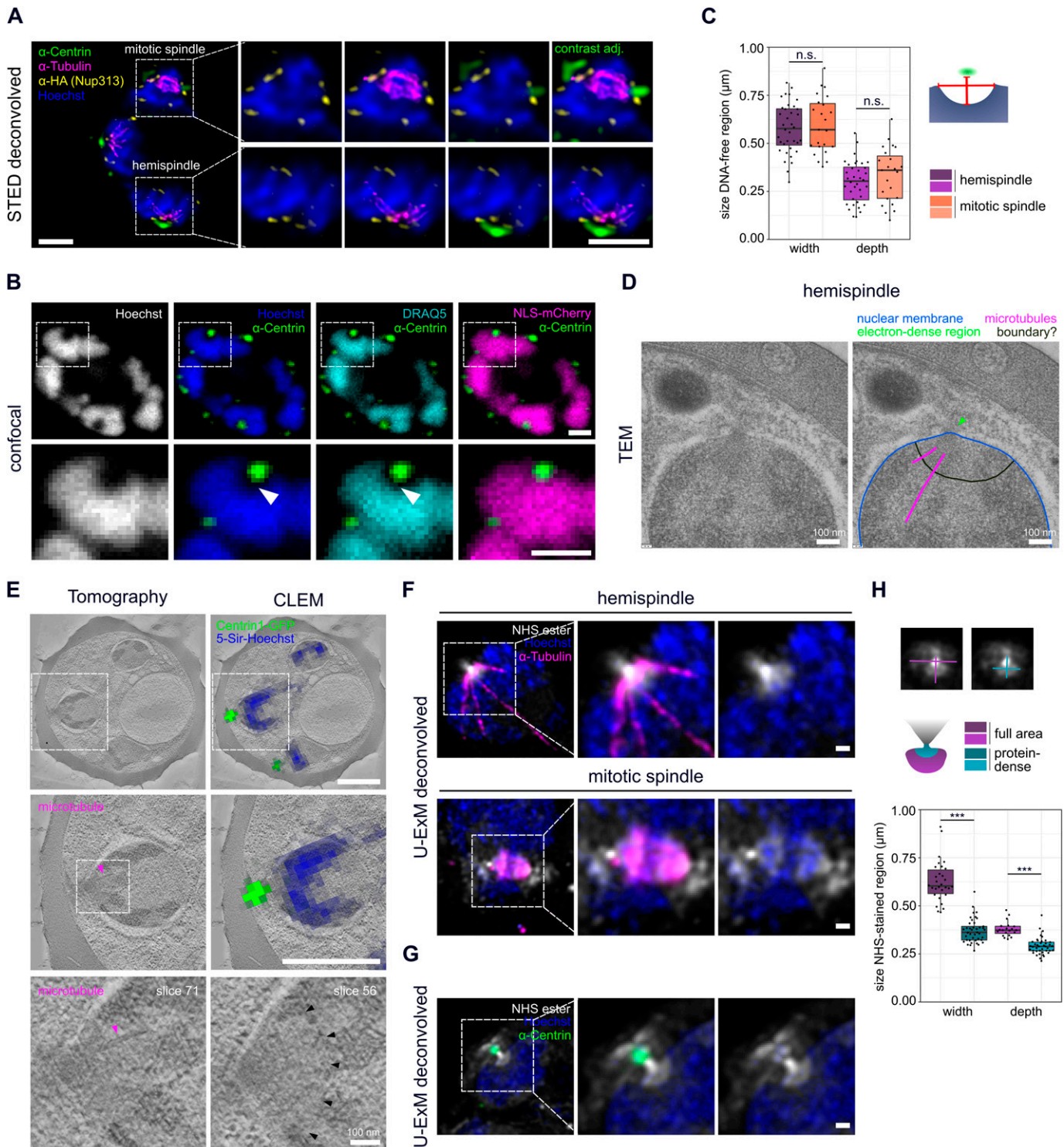

**Figure 3. Centriolar plaques are divided in an extranuclear centrin-containing compartment and an intranuclear DNA-free and protein-dense compartment associated with microtubules.**

**(A)** Dual-color STED nanoscopy images of a 3D7 schizont expressing tagged nuclear pore protein Nup313-3xHA_glms, labeled with anti-HA (yellow), anti-tubulin (magenta), and overlayed with confocal images of anti-centrin (green) and Hoechst staining (blue). **(B)** Confocal images of a 3D7 schizont ectopically expressing 3xNLS-mCherry. Signal was enhanced with RFP-Booster-Atto594 (magenta) and cells labeled with anti-centrin (green) and with Hoechst (blue) and DRAQ5 (turquoise) to detect DNA. **(C)** Quantification of dimensions of the DNA-free region in 3D7wt hemispindle (n = 36) and mitotic spindle (n = 23) stages using single image slices acquired in one immunofluorescence staining. Dimensions were measured as indicated in schematic. Depth was measured from underneath the centrin signal to the deepest point of the DNA-free region. Width was measured at the widest diameter of the DNA-free region where the nuclear membrane is expected. **(D)** Transmission EM image of the centriolar plaque region (annotated copy on the right) in a NF54 wt schizont shows no invagination of the nuclear membrane (blue) but suggests a boundary-like

intranuclear compartment with centrosomal function, which lacks chromatin but is dense in proteins.

Last, we wanted to improve on the structural preservation of microtubules in EM samples. For the CLEM approach (Fig 3E), we needed to embed cells in the resin LRgold, which can be polymerized chemically. Samples retrieved from UV polymerized HM20 showed superior contrast and were used for investigating more details of the microtubule organization in nuclei of the schizont stage by electron tomography. We found nuclear stages containing highly elongated individual microtubules which can deform the nuclear envelope at their tips and likely correspond to hemispindles (Fig 4A and Video 5). Here, the sample quality was sufficient to distinctly identify the characteristically shaped microtubule nucleation complex around γ-tubulin, which demarcates microtubule minus ends (Fig S6) (O'Toole et al, 2003; Höög et al, 2007). Those were emerging from discrete positions underlying an electron-dense region at the nuclear envelope. A distinct intranuclear compartment could, however, only be surmised in some nuclei when using this sample preparation method (Fig S7). Mitotic spindles displayed a short but much denser array of microtubules with minus ends clustered at a substantial distance from the nuclear envelope (Fig 4B and Video 6). Whereas this distance in hemispindles ranged from 26 to 96 nm, the distance for the mitotic spindle ranged from 88 to 204 nm (Fig S8). Taken together, these data suggest that intranuclear microtubule nucleation sites are embedded inside an extended amorphous matrix rather than linked to a membrane-associated centrosomal protein complex.

## Discussion

Our data provide an entirely novel perspective on the organization of the centriolar plaque and significantly expands on previous concepts depicting it merely as an electron-dense zone, which is inserted in the nuclear membrane (Prensier & Slomianny, 1986). This study highlights the significant differences to the well-characterized spindle pole bodies in budding yeast, where intranuclear microtubules emanate from a small and spatial well-defined protein complex which extends into the cytoplasm (Kilmartin, 2014). We reveal the organization of the centriolar plaque into an extranuclear and intranuclear compartment (Fig 4C). Contrary to what has been suggested (Mahajan et al, 2008), this study reveals that centrin is not embedded in the nuclear envelope but is part of a larger extranuclear compartment. Microtubule nucleation sites on the other hand are harbored by the intranuclear compartment, which contains a highly protein-

dense region. This organization discerns two potential regulatory hubs that could function in the coordination between the asynchronous division of the multiple nuclei in the cytoplasm and the chromosome segregation events within the nucleus. Although this seems to some degree reminiscent of the inner and outer core described for the centrosome in *T. gondii* (Suvorova et al, 2015), we note important differences. In *T. gondii*, it has been indicated that both cores are extranuclear, whereas centrin localizes to the outer core, where it is likely associated with the centrioles (White & Suvorova, 2018). They are likely separated by the prominent additional membrane layer that characterizes the *T. gondii* centrosome, but is absent from centriolar plaques. The centrocone, which is distinct from the inner core, is more reminiscent of the intranuclear compartment discovered here, but its relative positioning to the nuclear envelope or nuclear pores remains to be determined in *T. gondii*. It is characterized by the presence of MORN1, although a nucleus-associated MORN1 signal has not been described in *P. falciparum* (Ferguson et al, 2008; Rudlaff et al, 2019). Because of the consistent presence of nuclear pores at the centriolar plaque, it is plausible that they are involved in linking the intra- and extranuclear compartment. This is supported by the protein-dense structure uncovered here that stretches from the cytoplasm to the nucleoplasm and has a neck where it likely traverses the nuclear membrane. Whether those pores are specifically remodeled to constitute more "fenestra-like" structures remains to be investigated (Bannister et al, 2000).

We further clarify microtubule dynamics, which underlie their heterogeneous organization, previously described in fixed cells (Read et al, 1993; Fennell et al, 2008; Arnot et al, 2011), or using the microtubule-associated protein Kinesin-5 in live-cell imaging (Zeeshan et al, 2020a). Hemispindles are already present early in mononucleated cells indicating that they are not per se post-anaphase remnants as suggested earlier (Canning & Sinden, 1973). The lengths and numbers we measured are congruent with recent measurements (Bertiaux et al, 2021), although the slightly higher number of branches (5.7 versus 4) could be a consequence of analyzing structures in 3D instead of projections. Their function, however, remains elusive. We could abate the conspicuous hypothesis that they are directly involved in clustering centromeres at the nuclear periphery, which is consistent with findings in yeast where perinuclear centromere clustering has already been shown to be microtubule-independent (Richmond et al, 2013). An alternative explanation for the presence of hemispindles is that once tubulin accumulates beyond the required critical concentration microtubules polymerize spontaneously (Walker et al, 1988). Because appearance of the centrin signal is

structure (black) delineating an intranuclear region from which microtubules (magenta) emanate. Green arrow indicates electron-dense region likely associated with the centriolar plaque. **(E)** Correlative in-resin widefield fluorescence and electron tomography (CLEM) images of thick sections (300 nm) of a high-pressure frozen and embedded NF54 schizont expressing PfCentrin1-GFP (green) and stained for DNA with 5-SiR-Hoechst (blue). Same cell region containing clear PfCentrin1-GFP foci imaged by fluorescence microscopy was overlayed with an electron tomogram slice. In zoom-ins, arrows indicate a microtubule (magenta) and a boundary-like region (black) for two tomogram slices. **(F)** Confocal images of individual 3D7 Nup313-3xHA_glms schizont nuclei expanded with U-ExM in hemispindle and mitotic spindle phase. Proteins were labeled in bulk using an NHS-ester Atto594 dye conjugate (white). Brighter staining therefore indicates higher protein density. Cells were additionally stained with anti-tubulin (magenta) and Hoechst (blue). **(G)** as in (F), but cells were labeled with anti-centrin (green) instead of anti-tubulin. **(H)** Quantification of width and depth of the NHS conjugate-stained intranuclear region at the centriolar plaque for the full area as well as the highly protein-dense region as example image and schematic indicate. To test for significant differences, we used the Mann–Whitney U test. In total, we analyzed NHS-stained regions of 14 schizonts (one replicate).

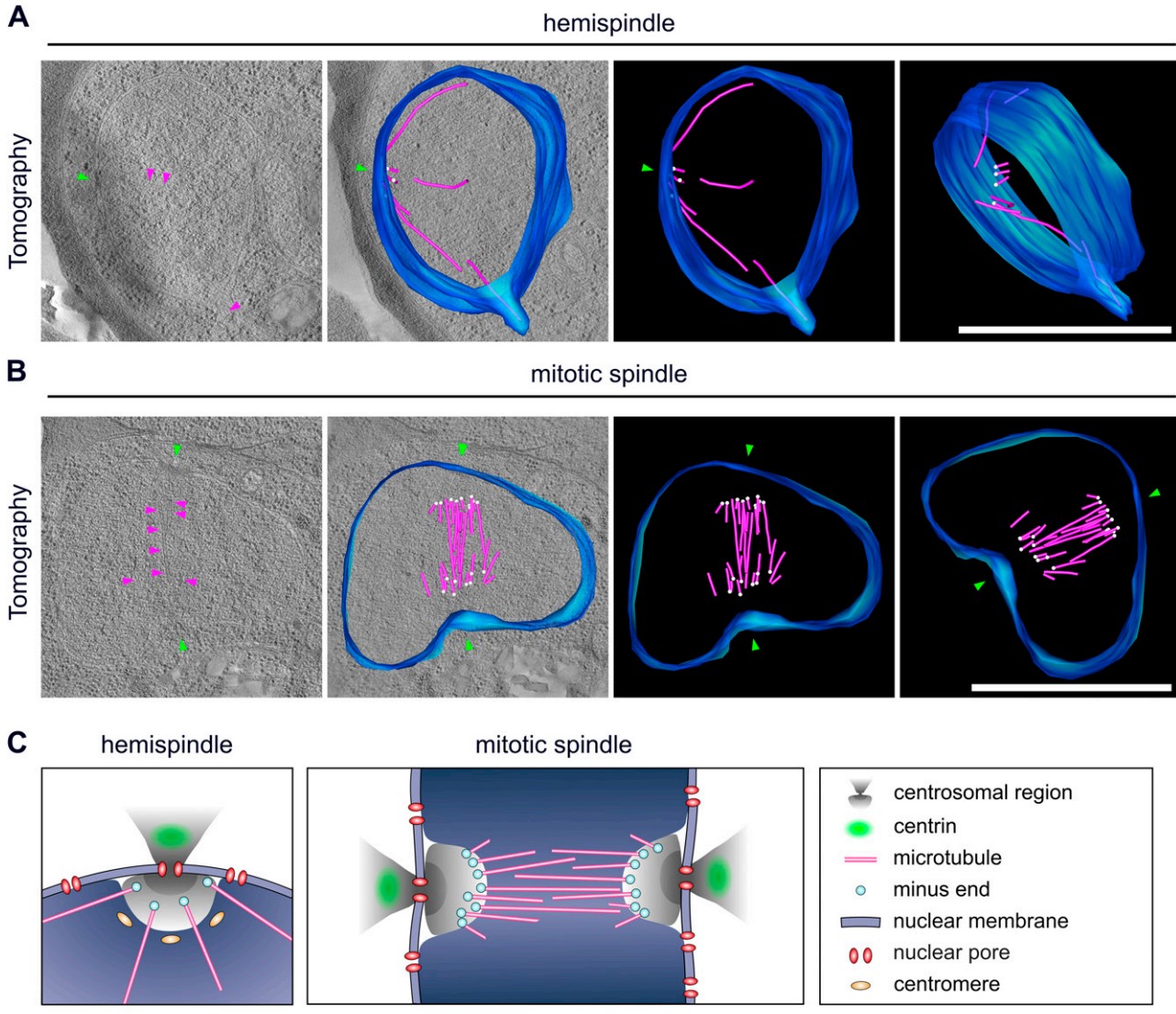

**Figure 4. Centriolar plaque microtubule nucleation sites are distinct and localize at a significant distance from the nuclear membrane.**
**(A)** 3D electron tomograms of thick sections (200 nm) of a schizont nucleus (NF54 PfCentrin1-GFP strain) in hemispindle stage. Corresponding surface rendering of microtubules (magenta), nuclear membrane (blue), microtubule minus ends (white), and electron-dense regions in the nuclear membrane (green) associated with the potential centriolar plaque are shown. **(B)** as (A) for mitotic spindle stage. All scale bars are 1 μm. **(C)** Schematic model of centriolar plaque organization during hemispindle and mitotic spindle phase in blood-stage schizonts. Content of the DNA-free, protein-rich intranuclear region harboring microtubule nucleation sites is unknown.

clearly delayed with respect to the hemispindle, we can speculate that centriolar plaque assembly occurs in a staged manner at the onset of schizogony. After the hemispindle stage, the tubulin signal collapses into a smaller focus, which is akin to the "tubulin-rich plaques" (Gerald et al, 2011). The increased sensitivity and resolution of our assays, however, demonstrate that mitotic spindles are virtually always associated with two centrin signals raising the hypothesis that those structures are early mitotic spindles. When and to which extent microtubules are polymerized into structured bipolar spindles is not resolved, but we can at this point exclude that they are the result of fusion between hemispindles, as previously suggested (Gerald et al, 2011). This early centriolar plaque duplication might be associated with S-phase onset as hypothesized earlier (Francia & Striepen, 2014).

Last, we did not observe intranuclear microtubules to be in contact with the nuclear membrane or membrane-associated structures, like for spindle pole bodies (Kilmartin, 2014). Rather, microtubules are nucleated from distinct sites within an extended nuclear compartment which is free of chromatin and has not been described before. In fission yeast, a similar configuration has been described although in this case meiotic microtubule arrays emanate from an amorphous compartment just outside the nucleus (Funaya et al, 2012). The displacement of minus ends away from the nuclear envelope upon mitotic spindle formation, whereas the dimensions of the centrosomal compartment stay constant, could be attributed to the force generated by the bipolar microtubule array. Despite the absence of centrioles, the preservation of the

nuclear membrane, and the lack of conserved factors, this structure is vaguely reminiscent of the pericentriolar material, which makes up the outer layer of vertebrate centrosomes and harbors microtubule nucleation complexes (Woodruff et al, 2014). This could indicate that this region might be occupied by a yet unknown matrix-like protein. Vertebrate centrosomes are duplicated by separation of the daughter and mother centriole, whereas yeast forms a new spindle pole body de novo. The amorphous and extended organization of the centriolar plaque revealed here, however, requires a new model to explain how this structure is duplicated or split upon formation of the two mitotic poles. Therefore, deciphering the composition of this novel nuclear compartment, understanding how it is assembled, and how it duplicates are some of the more pressing questions emerging from this study.

# Materials and Methods

## Parasite culture

*P. falciparum* cell lines NF54_Centrin1-GFP and 3D7_Nup313-HA_glms and NF54 wild type were cultured in O+ human red blood cells in RPMI 1640 medium supplemented with 0.2 mM hypoxanthine, 25 mM Hepes, 0.5% Albumax, and 12.5 μg/ml Gentamicin. Cultures were maintained at a hematocrit of about 3% and a parasitemia of 3–5%. Parasite cultures were incubated at 37°C with 90% humidity, 5% $O_2$, and 3% $CO_2$. To synchronize cultures, late-stage parasites were lysed by 5% sorbitol treatment.

## Plasmid constructs

To generate a pArl-Centrin1-GFP plasmid for episomal expression of PfCentrin1-GFP in NF54, we used a pArl-PfCentrin3-GFP plasmid (kindly provided by Tim Gilberger), which was generated based on the pArl backbone (Crabb et al, 2004). The pArl-PfCentrin3-GFP plasmid was digested with KpnI-HF and AvRII to cut out the PfCentrin3. The PfCentrin1 insert sequence was generated from *P. falciparum* cDNA via PCR using primers detailed in Table S1 alongside other used primers. PfCentrin1 and the digested backbone were ligated using Gibson Assembly and the sequence was verified by Sanger sequencing. To generate the pSLI-Nup313-3xHA_glms construct, first the HA and glmS sequences were ligated into the pSLI-TGD plasmid, a kind gift of Tobias Spielmann (Birnbaum et al, 2017), to obtain the pSLI-3xHA-glmS. The glmS ribozyme was amplified from the plasmid pARL_glmS (a kind gift from Jude Przyborski) and inserted by Gibson assembly into pSLI-TGD, downstream of the NeoR/KanR resistant cassette, using primers 0079 and 0080. For the 3xHA tagging, its sequence was first PCR amplified with primers 0126 and 0127 from pDC2-cam-coCas9-U6.2-hDHFR (a kind gift from Marcus Lee) (Lim et al, 2016) and cloned by Gibson assembly into the MluI and SalI digested pSLI-glmS plasmid. Last, 757 bp of NUP313 genomic sequence (without stop codon) was PCR amplified using primers 0163for and 0164rev and cloned into the modified pSLI-3xHA-glms plasmid using NotI and Mlu1 restriction sites. The p3-NLS-L3-mCherry construct was

obtained from the p3-NLS-FRB-mCherry plasmid, a kind gift of Tobias Spielmann (Birnbaum et al, 2017), by removing the FRB domain with NheI and KpnI digestion and consecutive ligation. Correct sequence of inserts was verified by Sanger sequencing.

## Parasite transfection

Transgenic parasites were generated by electroporation of sorbitol-synchronized ring-stage parasites with 50–100 μg of purified plasmid DNA (QIAGEN). To select for the plasmids pArl-PfCentrin1-GFP and p3-NLS-L3-mCherry, we used 2.5 nM WR99210 (Jacobus Pharmaceuticals) or 5 μg/ml blasticidin S (InVivoGen), respectively. To select for integration of the Nup313-3xHA_glms construct into the genome, we followed the protocol published previously (Birnbaum et al, 2017), using 800 μg/ml Geneticin-G418 (Thermo Fisher Scientific). PCRs across the integration junctions and testing for leftover unmodified locus to exclude the presence of wild type were performed (Fig S3 and Table S1). Limiting dilution was done to obtain clonal parasite lines.

## Seeding of infected red blood cells on imaging dishes

For live-cell imaging, cells were seeded on round imaging dishes with glass bottom (μ-Dish 35 mm, ibidi), for immunofluorescence staining on eight-well chambered glass slides (μ-Slide 8 Well, ibidi) as described previously (Gruring et al, 2011; Mehnert et al, 2019). Briefly, the glass surface was coated with Concanavalin A (Sigma-Aldrich, 5 mg/ml in water) for 20–30 min at 37°C. Dishes were rinsed twice with prewarmed incomplete RPMI 1640 medium lacking Albumax and hypoxanthine. Infected erythrocyte culture (500 μl for 35-mm dishes, 150 μl for each well of eight-well glass slides) was washed twice with incomplete medium by centrifugation (1,000*g*, 30 s), before addition of the cells onto the glass. Cells were allowed to settle for 10 min at 37°C. By gentle shaking and washing of the glass slides with prewarmed incomplete medium, unbound cells were removed until a monolayer of red blood cells remained on the glass surface. Incomplete medium was replaced by complete medium (4 ml for 35 mm dishes, 200 μl for each well of eight-well glass slides) and cells were maintained in the incubator until they were prepared for live-cell imaging or fixed for immunofluorescence staining.

## Immunofluorescence assay

Immunofluorescence staining for confocal and STED microscopy was performed as described previously (Mehnert et al, 2019). Briefly, after seeding cells at ring stage, parasites were fixed in schizont stages with prewarmed 4% PFA/PBS for 20 min at 37°C. PFA was washed off twice with PBS. Fixed cells were either stored in PBS at 4°C for later immunofluorescence staining, or stained immediately. First, the cells were permeabilized with 0.1% Triton X-100/PBS for 15 min at room temperature and rinsed three times with PBS. To quench free aldehyde groups, the cells were incubated with freshly prepared 0.1 mg/ml NaBH$_4$/PBS solution for 10 min. Cells were rinsed thrice with PBS and blocked with 3% BSA/PBS for 30 min. In the meantime, primary antibodies were diluted in 3% BSA/PBS and centrifuged at 21,100*g* for 10 min at 4°C to remove potential

aggregates. Cells were incubated with primary antibodies (Table S2) for 2 h at room temperature. Next, the cells were washed three times with 0.5% Tween-20/PBS. Incubation with secondary antibodies (Table S2) plus Hoechst in 3% BSA/PBS was performed for 1 h preceding removal of aggregates as described for primary antibodies. After washing twice with 0.5% Tween-20/PBS and once with PBS, cells were stored in PBS at 4°C in the dark until imaging. For longer storage, antibodies were occasionally fixed after staining with 4% PFA/PBS for 10–15 min at room temperature. After washing thrice with PBS, cells were likewise stored in PBS at 4°C in the dark.

### Antibodies

All antibodies and dyes used in this study are detailed in Table S3. Briefly, to stain microtubules, we used mouse anti-α-tubulin B-5-1-2. Polyclonal rabbit anti-TgCentrin1 antibody was a kind gift of Marc-Jan Gubbels and exclusively used for Fig 3B. Polyclonal rabbit anti-CenH3 antibody was a kind gift of Alan Cowman (Volz et al, 2010). To increase the signal of 3xNLS-mCherry (Fig 2B), cells were incubated with RFP-Booster nanobody coupled to Atto594 (ChromoTek) at a dilution of 1:200. To generate a polyclonal rabbit anti-PfCentrin3 antibody, a codon-optimized sequence of PfCentrin3 (PF3D7_1027700.1) was synthesized (Thermo Fisher Scientific, GeneArt Strings), and cloned into the pZE13d vector (Lutz & Bujard, 1997) with an N-terminal 6His tag via Gibson assembly (Hifi DNA Assembly, NEB) using ClaI and PstI restriction sites. The construct was transformed into chemically competent W3110Z1 *Escherichia coli* and colonies were inoculated in 800 ml LB-Amp expression culture, which was incubated at 37°C while shaking until an OD600 of 0.5 was reached. After induction with 1 mM IPTG, incubation continued for 3 h after which harvested bacteria were lysed via sonication. The lysate was cleared via centrifugation and recombinant PfCen3-6His were purified from the soluble fraction using Ni-NTA agarose beads (QiaGen) according to the manufacturer's recommendations. The buffer was exchanged to PBS via overnight dialysis and the protein further purified using the Superdex 75 10/300 size exclusion column (Cytiva). The final protein was used for a 63-d rabbit immunization regimen and affinity purification of the resulting serum performed by Davids-Biotechnology.

### Preparation of infected red blood cells for live-cell imaging

For live-cell imaging, NF54_PfCentrin1-GFP cells were seeded on glass bottom dishes as described above. Imaging medium, that is, phenol red-free RPMI 1640 supplemented with stable Glutamine and 2 g/l NaHCO3 (PAN Biotech) with all other supplements as in the parasite culture medium, was equilibrated in the cell culture incubator for several hours. Immediately before imaging, 9 ml of equilibrated imaging medium were supplemented with 4.5 µl (1:2,000 dilution) of the live microtubule dye SPY555-Tubulin (Spirochrome). List of all used dyes can be found in Table S3. Culture medium in the glass bottom dish with seeded cells was replaced by 8 ml imaging medium, the dish closed tightly without creating air bubbles, and sealed completely with parafilm. The imaging dish was directly taken to the incubation chamber of the microscope, prewarmed to 37°C.

### Super-resolution confocal and STED microscopy

Confocal microscopy of fixed and living cells was performed on a Leica TCS SP8 scanning confocal microscope with Lightning (LNG) module. LNG enables automated adaptive deconvolution after acquisition to generate super-resolution images. All images were acquired using a 63× 1.4 NA objective, GaAsP hybrid detectors and spectral emission filters. For live-cell imaging, the adaptive lightning acquisition mode was used with a pinhole of 1.2 airy units resulting in a pixel size of 53.8 nm and a total image size of 18.45 × 18.45 µm (344 × 344 pixels). The pixel dwell time was 488 ns. Every 5 min, a z-stack was taken of each cell with a total size of 6 µm and an z-interval of 0.5 µm. PfCentrin1-GFP was excited with a 488 nm laser at a laser power of 0.5%, SPY555-Tubulin was excited with a 561 nm laser at a laser power of 2%. Cells were imaged overnight for a maximum of 13 h. For confocal imaging of fixed cells, the LNG mode was turned off and images were acquired using a pinhole of 1 airy unit, a pixel size of 72.6 nm and a total image size of 9.3 × 9.3 µm (128 × 128 pixels). The pixel dwell time was 488 ns. Z-stacks of 6.27 µm were acquired with an z-interval of 0.3 µm. Rescue-STED microscopy was performed on a single-point scanning STED/RESOLFT super-resolution microscope (Abberior Instruments GmbH), equipped with a pulsed 775 nm STED depletion laser and three avalanche photodiodes for detection. Super-resolution images were acquired with a 100× 1.4 NA objective, a pixel size of 20 nm and a pixel dwell time of 10 µs. The STED laser power was set to 10–20%, whereas the other lasers (488, 594 and 640) were adjusted to the antibody combinations used. To prevent destruction of hemozoin-containing cells by the high-intensity STED laser, intensity thresholds (CONF levels) were defined, which needed to be reached in a confocal image before automatic activation of the STED laser (adaptive illumination). CONF levels varied between 10 and 110 and were adjusted individually for every cell. To acquire z-stacks (extended spindle, Fig 2A), a total z-stack of 3.9 µm was acquired using a z-step size of 300 nm.

### Ultrastructure expansion microscopy (U-ExM)

U-ExM was performed as described previously (Gambarotto et al, 2019, 2021), with slight modifications. Schizont parasite pellet was enriched using QuadroMACS Separator (Miltenyi) and added to a Poly-D-Lysin-coated coverslip to settle for 10 min at 37°C. Excess liquid was removed, and cells were fixed with prewarmed 4% PFA/PBS for 20 min at 37°C. Cells were washed three times with prewarmed PBS, and the coverslip was transferred into a six-well plate. After removal of PBS, the well was filled with 1 ml 1.4% Formaldehyde (Sigma-Aldrich)/2% Acrylamide (Sigma-Aldrich) in PBS and incubated for 5 h at 37°C. Sodium acrylate (7446-81-3; Santa Cruz Biotechnology) was solubilized in Milli-Q water. Protein denaturation was prolonged to 90 min at 95°C. The first round of expansion was performed in Milli-Q water for 30 min, before water was changed for overnight incubation. The gel was washed 2 × 15 min with PBS followed by blocking for 30 min with 3% BSA/PBS. We noted that to compensate for the reduced local protein concentration resulting from expansion, the use of multiple antibodies against different epitopes of tubulin increases the signal-to-noise ratio for microtubule staining as described previously (Gao et al,

2018). As primary antibodies, we used a combination of mouse anti-α-tubulin B-5-1-2 (T5168; Sigma–Aldrich), mouse anti-α-tubulin TAT-1 (00020911; Sigma–Aldrich), mouse anti-β-tubulin KMX-1 (MAB3408; Sigma–Aldrich), rabbit anti-PfCentrin3, and rat anti-HA 3F10 (12158167001; Sigma–Aldrich) diluted 1:250 in 1.2 ml 3% BSA/PBS. The solution was spun down for 10 min with 21,100$g$ at 4°C to remove aggregates. The gel was incubated with antibodies for 2 h 45 min at 37°C with agitation. Next, it was washed 5 × 10 min with 2 ml 0.5% Tween-20/PBS. As secondary antibodies, anti-mouse-STAR580 (Abberior) and anti-rabbit-Atto647 (Sigma–Aldrich) were diluted 1:100, and anti-Rat-Alexa488 (Thermo Fisher Scientific) was diluted 1:500 in 1.2 ml 3% BSA/PBS. Hoechst33342 (Thermo Fisher Scientific) was added in a dilution of 1:100, and the solution was spun down as previously described. Incubation was performed at 37°C for 2 h 30 min with agitation. The gel was washed 5 × 10 min with 2 ml 0.5% Tween-20/PBS afterward. For NHS-ester staining, the samples were additionally incubated with Atto594-NHS-Ester (ATTO-TEC, AD 594-31) at 10 $\mu$g/ml in PBS shaking for 1 h 30 min at RT and washed 3 × 15 min with 0.1% Tween-20/PBS. The second round of expansion was performed as described. We determined the expansion factor by measuring the gel with a ruler and found 4.5 on average. Samples were imaged on the Leica SP8 in standard confocal mode as described above, with a pixel size of 72.22 nm. Image analysis was performed as described above, and 3D movies were rendered using Imaris (Oxford Instruments).

## Image analysis and quantification

Images were analyzed using Fiji (Schindelin et al, 2012). Quantification of time-lapse images was performed on images after LNG adaptive deconvolution. Therefore, cells were examined manually to determine the changes of individual tubulin stages over time as well as the first stable appearance of the centrin signal. All deconvolved images shown were deconvolved using Huygens professional using express deconvolution with the standard template. Quantification of hemispindle and mitotic spindle length in U-ExM samples were measured using 3D distance measurement tools in Imaris and corrected by the expansion factor of 4.5×. Dimensions of DNA-free regions were measured in single slices of cells acquired with LNG mode on the Leica SP8, where the region underlying a centrin signal was visible from the side. Dimensions of NHS-stained protein-dense regions in expanded cells were measured in single slices after deconvolution of confocal images using the measurement tool in Fiji. Depiction of the measurement strategy is detailed in Fig 3H. 3D distances between MT ends and nuclear membrane were measured in the segmented tomography model using the mtk program in the IMOD software package. Data analysis and depiction were performed using Excel and R studio.

## Preparation of infected RBCs for electron tomography

For high-pressure freezing (HPF) of infected erythrocytes, late-stage parasites of the NF54_PfCentrin1-GFP strain (2–4 ml packed erythrocytes in culture, 3–5% parasitemia) were purified using magnetic activated cell sorting (VarioMACS Separator; Miltenyi Biotec). Importantly, schizonts were not in contact with PBS before

HPF, as we have shown recently that hemispindle microtubules are not detectable when parasites were fixed immediately after PBS incubation (Mehnert et al, 2019). For HPF, around 1.5 $\mu$l of concentrated purified schizont pellet was transferred into aluminum or gold carriers (3 mm diameter, 100 or 200 $\mu$m depth; Leica Microsystems) and high-pressure frozen with EM ice (Leica Microsystems). Freeze substitution was done in a Leica EM AFS2 (Leica Microsystems). Samples were freeze-substituted in 0.3% uranyl acetate in dry acetone for 24 h at −90°C, followed by an increase in temperature from −90°C to −45°C in 9 h (5°C/h). Samples were incubated for another 5 h at −45°C, before rinsing 3 × 10 min with dry acetone. Acetone was replaced by increasing concentrations of the Lowicryl HM20 (25%, 50%, and 75%) in dry acetone at −45°C for 2 h each. Cells were incubated in 100% HM20 at −45°C, after 12 h the solution was again replaced by 100% HM20 and incubated for another 2 h at −45°C. To polymerize HM20 and therefore embed the samples in the resin, UV light was applied for 48 h at −45°C, for another 13 h, while increasing the temperature from −45°C to +20°C (5°C/h) and 48 h at 20°C. The RBC pellets were not well polymerized because of the pigmentation of the cells and areas with well-embedded cells had to be selected for sectioning. Polymerized cells were trimmed and sectioned on a UC7 ultramicrotome (Leica Microsystems). 200 nm-thick sections were collected on Formvar-coated copper slot grids and contrasted with 3% uranyl acetate and Reynold's lead citrate. Sample quality was checked on a Jeol JEM-1400 80 kV transmission electron microscope equipped with a 4k by 4k pixel TemCam F416 digital camera (TVIPS). For image acquisition, the EM-Menu (TVIPS) software was used. For tomography, sections were placed in a high-tilt holder (Model 2040; Fischione Instruments) and the cells were recorded on a Tecnai F20 EM (FEI) operating at 200 kV using the SerialEM software package (Mastronarde, 2005). Images were taken every degree over a ±60° range on an FEI Eagle 4K × 4K CCD camera at a magnification of 19,000× and a binning of two (pixel size 1.13 nm). The tilted images were aligned using tilt series patch tracking. The tomograms were generated using the R-weighted back-projection algorithm. To reconstruct the complete hemispindle, tomograms were collected from three serial sections, aligned, and joined by using the eTomo graphical user interphase (Höög et al, 2007). Tomograms were displayed as slices of one voxel thick, modelled, and analyzed with the IMOD software package (Kremer et al, 1996). Capped ends of microtubules were identified as minus ends in accordance to earlier microtubule studies with similar preservation techniques (O'Toole et al, 2003; Höög et al, 2007; Gibeaux et al, 2012).

## Correlative in-resin widefield fluorescence and electron tomography (CLEM)

For in-resin CLEM, magnetically purified NF54_PfCentrin1-GFP late-stage parasites were incubated with 1 $\mu$M of the live dye 5-SiR-Hoechst for about 1 h at 37°C. HPF was performed as described above. Freeze substitution and embedding were carried out in an Automatic Freeze Substitution System (AFS2; Leica Microsystems), but pipetting steps were performed manually. Cells were freeze-substituted in 0.3% uranyl acetate in dry acetone for 29 h at −90°C, before the temperature was increased to −45°C in 9 h (5°C/h) and

kept at –45°C for at least 5 h. The freeze-substitution solution was replaced by 100% cold, dry ethanol and the temperature was increased from –45°C to –25°C in 1 h (20°C/h). Because of difficulties using UV for resin polymerization in the pigmented erythrocytes, samples were incubated with increasing concentrations (25%, 50% and 75%) of LRGold (London Resin company) in dry ethanol at –25°C for 2 h each. Afterward, 100% LRGold was added, removed, and again added for overnight incubation of the samples at –25°C. The resin LRGold was supplemented with the initiator, 1.5% benzoyl peroxide, on ice. The solution was inverted carefully to prevent oxygen incorporation and immediately placed at –20° to prevent direct polymerization. Samples were incubated with 100% LRGold with initiator for 26 h at –25°C. Temperature was increased from –25°C to 20°C in 9 h (5°C/h) and samples stayed at 20°C for 24 h for full polymerization. 300 nm-thick sections were cut using a UC7 ultramicrotome (Leica Microsystems) and collected on Formvar-coated finder grids. Immediately after sectioning, the grid was placed in a drop of PBS with pH 8.4 (Ader & Kukulski, 2017) on a glass coverslip with the sections facing the bottom. A second glass coverslip was added on top and the sandwich was mounted in a metal ring holder (Kukulski et al, 2011). Fluorescence was imaged directly on a Zeiss Axio Observer.Z1 widefield system equipped with an AxioCam MR R3 camera and a 63× oil objective (1.4 numerical aperture). For excitation of PfCentrin1-GFP, a 488-nm laser was used, for excitation of 5-Sir-Hoechst, a 587-nm laser, both set to a laser power of 95% and an exposure time of 900 ms. Images were taken with a pixel size of 102 nm and a total size of 1,388 × 1,040 pixels per image. Subsequently, the sections were contrasted with 3% uranyl acetate and Reynold's lead citrate. After checking the ultrastructure quality of the PfCentrin1-GFP positive cells on a Jeol JEM-1400 80 kV transmission electron microscope (Jeol), tomography was performed on a Tecnai F30 EM TEM (EMBL Heidelberg), operating at 300 kV. Tilt series of 300 nm-thick sections were recorded at the range of ±60° with 2° interval, on a 4x4k CCD camera (Gatan - OneView), using SerialEM acquisition software. 3D reconstructions and further analyses were conducted using "etomo" Image Processing Package. Correlation of fluorescence and electron tomography images was performed manually using Fiji, GIMP 2.10.20., and Inkscape.

### Transmission EM of Spurr-embedded infected RBCs

For HPF, erythrocytes infected with NF54 wild-type parasites (500 µl packed red blood cells in culture, 6% parasitemia) were purified using magnetic activated cell sorting (QuadroMACS; Miltenyi Biotec). Purified late stages were accompanied with 30 µl uninfected red blood cells and cultured for 4 h to let the cells recover. High-pressure freezing using Leica EM ICE (Leica Microsystems) was performed as described above. Freeze substitution was done in an Automatic Freeze Substitution System (AFS2; Leica Microsystems). Samples were freeze-substituted in 0.2% Osmium tetroxide, 0.3% uranyl acetate and 5% $H_2O$ in dry acetone for 1 h at –90°C. Temperature was increased from –90°C to +20°C in 22 h (5°C/h) and samples stayed at 20°C until further processing. Next, samples were washed three times with dry acetone. The pellets detached from the carriers and were combined in a 1.5 ml reaction tube in dry acetone and pelleted for 2 min, 325$g$. Acetone was replaced by a 1:1

mixture of Spurr's resin (Serva) and dry acetone. After 2 h incubation at room temperature, the mixture was replaced by 100% Spurr's resin and incubated overnight at room temperature. The Spurrr's resin was removed and again replaced by 100% Spurr's resin. Polymerization of the samples was performed at 60°C for 1–2 d. Samples were trimmed with a UC7 ultramicrotome (Leica Microsystems) and 70 nm thin sections collected on Formvar-coated slot grids. Images were taken on a Jeol JEM-1400 80 kV transmission electron microscope (Jeol) equipped with a 4k by 4k pixel TemCam F416 digital camera (TVIPS). For image acquisition, the EM-Menu (TVIPS) software was used.

## Supplementary Information

## Conflict of Interest Statement

The authors declare that they have no conflict of interest.

## Acknowledgements

We thank the Infectious Diseases Imaging Platform for imaging support (idip-heidelberg.org), The Electron Microscopy Core Facility at Heidelberg University for providing electron microscopy services, PlasmoDB for their *Plasmodium* Informatics Resources, Stefan Pitsch and Luc Reymond (Spirochrome LTD) for providing their SPY555-Tubulin dye, Grazvydas Lukinavicius and Jonas Bucevicius (Max Planck Institute for Biophysical Chemistry) for providing the 5-SiR-Hoechst dye, Paul Guichard (University of Geneva) for sharing their U-ExM protocol, Nicolas Lichti for helping with molecular cloning, Marc-Jan Gubbels (Boston College) for the anti-TgCentrin1, and Alan Cowman (WEHI Melbourne) for the anti-CenH3 antibody. We thank the German Research Foundation (DFG) (349355339), the Human Frontiers Science Program (CDA00013/2018-C), and the Daimler and Benz Foundation for providing funds to J Guizetti and the "Studienstiftung des Deutschen Volkes" to Y Voβ; the German Research Foundation (DFG)—project number 240245660—SFB 1129 for providing funds to M Ganter and D Klaschka; the Baden-Württemberg Foundation (1.16101.17) for providing funds to M Ganter; and the Fundação para a Ciência e Tecnologia (FCT, Portugal)—PD/BD/128002/2016 for providing funds to M Machado.

### Author Contributions

CS Simon: conceptualization, resources, data curation, formal analysis, validation, investigation, visualization, methodology, and writing—original draft, review, and editing.
C Funaya: investigation, visualization, methodology, and electron microscopy.
J Bauer: investigation, visualization, methodology, and expansion microscopy.
Y Voβ: resources, methodology, and centrin antibody generation.
M Machado: resources, visualization, Nup313-HA strain, and NLS-mCherry strain.
A Penning: resources.
D Klaschka: resources.

M Cyrklaff: investigation, visualization, and parts of electron tomography.

K Kim: investigation, visualization, and parts of electron tomography.

M Ganter: resources, supervision, and funding acquisition.

J Guizetti: conceptualization, data curation, supervision, funding acquisition, investigation, visualization, methodology, project administration, and writing—original draft, review, and editing.

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
