## [Reviewer comments · Life Science Alliance]

Life Science Alliance

An extended DNA-free intranuclear compartment organizes centrosome microtubules in malaria parasites

Caroline Simon, Charlotta Funaya, Johanna Bauer, Yannik Voß, Marta Machado, Alexander Penning, Darius Klaschka, Marek Cyrklaff, Juyeop Kim, Markus Ganter, and Julien Guizetti

DOI: <https://doi.org/10.26508/lsa.202101199>

Corresponding author(s): Julien Guizetti, Centre for Infectious Diseases, Heidelberg University Hospital

Review Timeline:

Submission Date:	2021-08-18
Editorial Decision:	2021-08-24
Revision Received:	2021-08-31
Accepted:	2021-09-03

Scientific Editor: Novella Guidi

Transaction Report:

Please note that the manuscript was reviewed at Review Commons and these reports were taken into account in the decision-making process at Life Science Alliance.

Review
COMMONS

Authors' Response to Reviewers Comments

Dear reviewers,

We thank all reviewers for and their appreciation of our work and even more so for their constructive comments and suggestions, which will significantly improve the quality of the manuscript. We were able to complete the revision and address all reviewer comments. Aside a more stringent discussion of the literature, and rewording of certain paragraphs for clarity, we also generated additional experimental data.

More importantly, to address the concern that we did not provide a positive marker for the intranuclear compartment, we present new images. We attempted to label gamma-Tubulin by generating new antibodies, GFP-tagged strains, and trying multiple commercial antibodies since the beginning of the project. Only recently we found an antibody providing a more specific signal at the expected location, although with some likely cross-reactivity with alpha- and beta-tubulin, and now show these data in the supplements. Additionally, we generated expansion microscopy samples stained with a fluorophore-coupled NHS-Ester, a bulk protein label. These data show that the centrosome contains an exceptionally protein dense hourglass-shaped region, which spans from the extranuclear to the intranuclear compartment, as revealed by centrin and tubulin co-staining. This fortifies our claims about the distinct nature of the intranuclear centrosome compartment containing the microtubule nucleation sites.

Further, we add images of 5-SiR-Hoechst, SPY555-Tubulin, Centrin1-GFP triple labelling live cells to demonstrate the specificity of the microtubule dye and to underline that we are indeed acquiring the dynamics from the first nuclear division on.

In terms of formatting we added line numbers and uploaded high quality figures separately. Due to the added data and panels we needed to split Fig. 1 into two separate figures, rewrote the figure legends and moved them to the end of the document.

Please find below a point-by-point response to the comments.

Best regards,
Julien Guizetti

Reviewer #1 (Evidence, reproducibility and clarity (Required)):

The manuscript by Simon and collaborators addresses the dynamic changes of spindle and hemispindle microtubules occurring along schizogony in *Plasmodium falciparum*. The work explores the temporal correlation of the changes observed in intranuclear spindles with changes at the level of the centriolar plaque; the nuclear microtubule organizing center of these parasites, using centrin as a bona fide marker of the structure. The study shows that spindle microtubules organize from an intranuclear region, devoid of chromatin, distinct from the centrin region which had not been observed or described before. It further shows that centrin does not localize at the nuclear envelope, but it is actually extranuclear.

This work significantly expands on previous knowledge regarding the functional and spatial organization of the nucleus in *P. falciparum*, and the structure once defined as "an electron dense mass

on the nuclear envelope." It uses state of the art microscopy approaches such as STED, UExM and CLEM, in combination with immunolabeling, dyes and parasites over expressing fluorescent protein fusions, to address these questions.

****Major comments:****

- Are the key conclusions convincing?

I find the manuscript successfully addresses the posed questions. The data presented supports the conclusions.

- Should the authors qualify some of their claims as preliminary or speculative, or remove them altogether?

- Would additional experiments be essential to support the claims of the paper? Request additional experiments only where necessary for the paper as it is, and do not ask authors to open new lines of experimentation.

No

- Are the suggested experiments realistic in terms of time and resources? It would help if you could add an estimated cost and time investment for substantial experiments.

N/A

- Are the data and the methods presented in such a way that they can be reproduced?

Yes

- Are the experiments adequately replicated and statistical analysis adequate?

Yes

****Minor comments:****

- Specific experimental issues that are easily addressable.

On the data shown in Figure 1, it is unclear to me what elements are taken into account to define "anaphase." Anaphase could be defined by using chromatin markers - such as CenH3- which have been identified in Plasmodium and the authors make use of in Figure 1F.

We acknowledge that the term anaphase is ill-defined here. Further it suggests a mitotic morphology analogous to the one observed in "classical" models (prophase, metaphase, anaphase,...), which is not fully appropriate. In line with the comments by Reviewer 3 we, therefore, decided to use the term "extended spindle" instead (Fig. 1 & 2). This better reflects the morphological criterion on which we based the stage definition.

- Are prior studies referenced appropriately?

The authors state that "with the exception of centrins and gamma tubulins" few canonical centrosome components are conserved in Plasmodium. These parasites are in fact able to assemble a more or less canonical centriole for microgamete basal body formation. Widely conserved centriolar components such as Sas6 are coded by the malaria genome, and have been characterized previously. This work is neither referenced nor discussed in the manuscript.

The reviewer is right to point out this omission. We were too much focussed on the blood stage centriolar plaques while writing this section, where centrioles are not observed. Of course centriole-

like structures are relevant in other life cycle stages, such as microgametes, and should be discussed (line 104). Some previous attempts to endogenously tag Sas6 to verify its localization in blood stages were unfortunately not successful.

- Are the text and figures clear and accurate?

I find the timings shown in Figure 1A, with respect to the schematic quantification shown in Figure 1B, confusing. Shown as it is, one naturally correlates the images on Fig1A above with the cell cycle progression timing shown on Fig1B, below. However, by time 260min, for example, two somewhat adjacent centrin signals can be observed. Though this is defined as anaphase- by an unspecified criterium- this could very well be representative of metaphase. Nonetheless, the timing shown on Figure 1B for "anaphase" onset is 170min, which is inconsistent with the images above. I suggest that either, the quantification is shown in a different format (ex. bar plots) which could then better reflect the cell to cell variations observed (by use of error bars, for example) or that the figure explanation in the results section clarifies this issue.

We understand how this representation is misleading and have adjusted the figure and text accordingly. We modified the time stamps in Fig. 1A (now Fig. 1C) to the scale used in Fig. 1B (now Fig. 1D) i.e. collapse of the hemispindle is $t=0$ and explain this in the text (line 158). Since we feel that Fig. 1B (now Fig. 1D) is a good and compact visual representation of progression through the first division we kept the bar plots in the supplements (Fig. S1), but added a title clarifying that average duration between multiple movies are shown.

As presented, the data in Figure 1C is rather uninformative. A pattern could be more immediately extracted if dots corresponding to subsequent appearance of centrin dots in the same nucleus were connected to each other.

Concerning the appearance of the centrin signals we adopted the good suggestion by the reviewer and connected "paired" centrin signals by lines (Fig. 1E).

- Do you have suggestions that would help the authors improve the presentation of their data and conclusions?

There are a number of edits required on the text. Row numbers would have been helpful in pointing these out. I point some edits below, but thorough revision of the manuscript for grammatical and synthetic errors would be beneficial.

- Cytokinetic segmeter - please replace with "segmented"
- Please refer to Figure 1D when appropriate - there is quite an extensive paragraph describing the results shown on this figure, but it is only referenced at the start.
- "..., as did the and the number of branches per nucleus,..." please rewrite as appropriate.

We apologize for not providing line numbers, but have corrected the addressed points and applied a grammatical check throughout the manuscript. We have added additional references to Figure 1D (now Fig. 2A) in the text.

Reviewer #1 (Significance (Required)):

This manuscript could be interesting to a wide audience interested in cell cycle, cell division, cell organization and organelle positioning, infectious diseases and microscopy. However, the introduction assumes that readers are somewhat experts in the malaria field. I suggest the authors include a brief introduction of the malaria life cycle, and a schematic representation of the division mode. This will help non-experts follow the narrative more easily.

We are happy to read that the reviewer sees value of this study for a broader audience. Following the suggestion, we added a small schematic (Fig. 1A, lines 54, 62) highlighting the relevant steps of schizogony and expanded the introduction of the life cycle (line 46).

This work rectifies long-standing inconsistencies observed by different experimental approaches in the nuclear organization of malaria parasites during schizogony. However, what the functional consequences of the alternative modes of spindle organization in malaria could be, are not clearly stated or discussed. In this respect, as it stands, the manuscript is rather descriptive and lacks mechanistic insight. Nonetheless, the data presented are of superb quality, and the manuscript represents a tremendous leap in structural insight and imaging resolution for the field of malaria. I find the data is suitable for publication albeit minor adjustments are made (specially to Figure 1 and/or the description of the results shown in Figure 1, for consistency).

We agree that the value of this manuscript lies in the clarification of conflicting data, unprecedented structural insight, and providing a useful working model for the malaria parasite centrosome. Although this study is ultimately descriptive it forms the indispensable basis to generate more meaningful functional insight about centrosome biology and nuclear division. Some of the functional consequences worth considering are: i) The (at least) bipartite composition indicating that centrosome functionality is spatially spread throughout the nucleoplasm/cytoplasm boundary. ii) The delayed appearance of the centrin signal after tubulin signal allows the prediction that centrosome assembly is a staged process occurring over an elongated period of time. iii) The generally amorphous structure of the compartment predicts the involvement of yet to be uncovered matrix-like proteins harbouring microtubule nucleation sites. iv) Lastly, our model has important implications for the mechanism of centrosome duplication. In a centrosome containing centrioles (like in vertebrates), the duplication event can easily be explained by physical separation of the daughter and mother centrioles. Spindle pole body duplication in yeasts is achieved by de novo formation of a new one, which remains connected by a half bridge until it is split. The centriolar plaque organization revealed here suggests that we need an entirely new model of centrosome duplication (or splitting) to describe and understand this process in malaria parasites. We now address those points more explicitly in the discussion section (e.g. lines 375, 443, 467).

****Referee Cross-commenting****

I agree with all the other reviewer's comments. I'm glad the reviewers seem to be experts in the field of malaria cell division and have pointed out previous studies which were not appropriately referenced. I second those comments.

Reviewer #2 (Evidence, reproducibility and clarity (Required)):

****Summary:****

The manuscript by Simon et al have used advance cell biology technology like STED, expansion and live cell imaging to decipher the configuration of microtubules, centrin and nuclear pore during unconventional cell division process in malaria parasite. They have shown the dynamics of centrin and its localisation with respect to centriolar plaque that is characteristic of these parasite cell during schizogony> They also implicate from their studies that there is extended intranuclear compartment which is devoid of chromatin

****Major Comments****

- Are the key conclusions convincing? Yes to some extent
- Should the authors qualify some of their claims as preliminary or speculative, or remove them altogether?

*Some part are preliminary and speculative as there is no solid data supporting it. Please see below

- Would additional experiments be essential to support the claims of the paper? Request additional experiments only where necessary for the paper as it is, and do not ask authors to open new lines of experimentation.

*Yes to substantiate their claim

- Are the suggested experiments realistic in terms of time and resources? It would help if you could add an estimated cost and time investment for substantial experiments.

*They can do these quite quickly less than a month

- Are the data and the methods presented in such a way that they can be reproduced?

*Yes

- Are the experiments adequately replicated and statistical analysis adequate?

*Yes

The authors present beautiful imaging and some in depth structure using tomography and CLEM to show the location of centrin which is generally considered the marker for centrosome or Microtubule organising centre in malaria parasite. These approaches are still not been applied in Plasmodium and hence very informative. Though they present some advance microscopy but a lot of these concept for hemispindle were shown earlier in many light and super resolution microscopy studies. Authors claim that they are first to show that there is space between centrin and nucleus but it has been show previously in centrin studies in Plasmodium berghei using super resolution microscopy (Roques et al 2019 Fig1 and supplementary videos1&2) as well as expansion microscopy recently by group of Brochet etal 2021.

We thank the reviewer for the appreciation of our work. We are, indeed, not the first to describe the gap between centrin and tubulin or the nucleus. We just aimed to reiterate this finding, also visible in our data, in order to transition to the analysis of nuclear pore positioning to clarify whether centrin is

actually extranuclear. Nevertheless, we should have cited the Roques and Bertiaux et al. studies again in this context, which we have now rectified (line 252).

In addition the microtubule dynamics was also recently shown with Kinesin5 live cell imaging for schizogony in *Plasmodium berghei* (PMID: 33154955) which author have omitted in their manuscript. We thank the reviewer for pointing out the Kinesin-5 study by Zeeshan et al., which we failed to cite and discuss. We now state the findings of this publication and put it into the context of our work (see also answer to next point). Microtubule associated proteins, such as the microtubule plus end tracking EB1 and the aforementioned Kinesin-5, are indeed useful markers to investigate microtubule dynamics leading to the interesting results shown by Zeeshan et al. Nevertheless, we want to point out that labelling microtubule associated proteins (MAPs) remains an approximation of the underlying microtubule organization. As the authors in Zeeshan et al. indicate by themselves, Kinesin-5 does not decorate axonemal microtubules or the membrane-associated microtubule structure formed during cytokinesis in very late schizont stages. Further, colocalization between alpha-tubulin and kinesin-5 in schizont-stage parasites is not complete indicating a preferential decoration of certain sections of the microtubule structures (possibly the microtubule ends), which could only be resolved by super-resolution microscopy. Using a live cell dye, such as SPY555-tubulin, which directly binds to microtubules will provide a uniform labelling of any microtubule species and hopefully prove useful to the field in the future. Lastly, we present time-lapse microscopy analysis of blood stage cells, contrary to single time point images of live cells, providing a quantified chronology of microtubule reorganization at single cell level (with time stamps). Therefore, we feel that our claim, although it should be relativized, is formally speaking accurate.

It is also important that authors give valid discussion about previous studies on hemispindle, microtubule dynamics with respect to schizogony (PMID: 18693242; PMID: 11606229; PMID: 33154955) rather than giving the impression that they have given this concept first time on hemispindle dynamics and centrin location during schizogony.

We agree that those studies should be discussed in more detail. We are grateful to the reviewer for pointing out the Fowler et al. 2001 (PMID: 11606229) study. They use an antibody against gamma-tubulin to demonstrate its presence at the apical pole of subpellicular microtubules (f-MAST) in the merozoite and cytokinetic stages (line 102). However, we were unable to reveal a specific gamma-tubulin staining using the antibody used by them in the preceding schizont stage. After trying many different commercial gamma-tubulin antibodies and attempting to generate our own we now finally observe a gamma tubulin localization at the poles of intranuclear spindles in schizont stage, although the only successful antibody still displays some background staining, possibly including cross-reactivity with alpha or beta-tubulin (Fig. S4, line 237).

The highly insightful study by Mahajan et al. 2008 (PMID: 18693242) indeed suggests that centrin localizes away from the DNA and demonstrate the distinct localization from tubulin. They, however, likely due to the resolution limit of their microscopy techniques, speculate that the centrin signal is embedded in the membrane, while we could show by super-resolution and nuclear pore staining that centrin is distinct from the membrane (now Fig. 2A; line 257). The work done by Zeeshan et al. 2020

(PMID: 33154955) nicely shows dynamics of kinesin-5 in nuclear division. In schizont stages Kinesin-5 signal elongates and splits alongside the mitotic spindle with which it overlaps for the most part. Colocalization with centrin is less strong although the authors note some overlap. Our data suggest that centrin and tubulin are clearly distinct. In male gametes the authors show nicely time-resolved data of kinesin spreading along the elongating spindle, although hemispindles are not observed at this stage. We introduce and discuss these findings (lines 123, 432).

The concept of bipartite centrosome is already been discussed in *Toxoplasma* and the claim by authors in *Plasmodium* presented here is not substantiated experimentally. They showed that centrin is part of outer region while they do not show with any marker for the inner region. It will be very helpful if the authors use gamma tubulin or MORN1 to show the location with respect to centrin and microtubule. In the absence of this localisation the claims are preliminary and speculative. If the centrosomal protein complex is not involved in microtubule nucleation, then how the nucleation is happening. What are the molecules present in this amorphous matrix? It will be great to check the location of gamma-tubulin or some inner centrosome molecules described in *Toxoplasma* that is deemed to be MTOC.

We share the opinion that our *Plasmodium* data should be compared to *Toxoplasma*, while still being assessed independently. Despite *Toxoplasma* belonging to the apicomplexan the conclusion that their centrosomes should be organized in a similar fashion is by no means self-evident considering for example their significant evolutionary distance. Actually, several noteworthy morphological differences have already been well documented. i) *Toxoplasma* MTOC does contain centrioles in the outer core which is coherent with the centrin and gamma-tubulin localization in this region. ii) *Toxoplasma* MTOC contains an additional nuclear membrane protrusion enclosing the inner core. iii) mitotic microtubules in *Toxoplasma* are thought to penetrate the nuclear membrane to connect to centromeres. iv) the inner and the outer core are both extranuclear and therefore not to be equated with the intranuclear compartments. We now expand a bit on the discussion of the aforementioned differences (line 382). Nevertheless, we thank the reviewer for making us realize that the term “bipartite” is a poor choice to describe the centriolar plaque organization in this context. Therefore, we replaced it in the abstract (line 29) and the main text (line 375).

We acknowledge the fact that it would be desirable to show a marker localizing to the intranuclear compartment, and not only through visualizing the microtubule nucleation complex (Fig. 4A-B) and the positioning of the microtubule ends in this region (Fig. 3A). Concerning MORN1 we found no indication in the published localization data that it is, like in *Toxoplasma*, associated with the nucleus in *Plasmodium* species, where it is only found associated with the budding complex (and we are currently unable to procure an antibody) (line 422). We have attempted gamma-tubulin visualization on many occasions throughout the project (transgenic parasite lines, commercial antibodies, self-made antibodies) and only recently found an antibody revealing some specific signal. Indeed, we found localization at the poles of the spindles i.e. the intranuclear compartment (line 237). Unfortunately, this “best-possible staining” still showed some unspecific spindle staining likely resulting from cross-reactivity with alpha- or beta-tubulin causing us to put these data into the supplements (Fig. S4).

We had more luck with attempting a “new” type of staining, recently used in Plasmodium (Bertiaux & Balestra et al. 2021) using a fluorophore-coupled NHS-Ester in expanded samples. This chemical unspecifically stains proteins and revealed that the centrosomal region contains an exceptionally protein dense “hourglass-shaped” structure (Fig. 3F-H). Since the outer part of this structure colocalizes with centrin and the inner part overlaps with microtubules we assume that the centrosomal complex stretches throughout the nucleocytoplasmic boundary and fills part of the intranuclear compartment (line 320). Especially the highly protein dense region at the neck of the “hourglass” seems very coherent with the nuclear membrane embedded electron dense region which can be seen in electron microscopy (e.g. Fig. 3E & 4B). We feel that this staining strongly supports the presence of this novel intranuclear compartment.

The expansion microscopy is very nice and some of it presented in supplementary can be moved to main section.

Thanks for sharing our enthusiasm about this imaging technique. We have now selected a representative image of a hemispindle and mitotic spindle stage nucleus imaged by U-ExM and added it to the main section (Fig. 2B, line 231).

The localisation CenH3 is bit puzzling as it has been shown that centromere/ kinetochore cluster and are present during early and mid schizogony. The various foci with respect to nuclei are not what has been seen previously. Please discuss the difference in these two findings.

The localization pattern can easily be explained by the increased resolution of STED nanoscopy used in this study. Previous studies (e.g. Hoeijmakers et al. 2012 and Zeeshan et al. 2020) used classical confocal microscopy. Under those imaging conditions the individual foci seen here can't be resolved and would, in accordance with the other studies, appear as one cluster. We slightly modified the text for more clarity (line 247).

Reviewer #2 (Significance (Required)):

- Describe the nature and significance of the advance (e.g. conceptual, technical, clinical) for the field.
- * This is more technical advancement on the subject of centrin by using STED, tomography and CLEM.
- Place the work in the context of the existing literature (provide references, where appropriate).
- * This work has relevance relation to cell division during schizogony in asexual stages in par with Toxoplasma or in Apicomplexa in general
- State what audience might be interested in and influenced by the reported findings.

Working with Apicomplexa, Protist, cell division and mitosis.

- Define your field of expertise with a few keywords to help the authors contextualize your point of view.
- Indicate if there are any parts of the paper that you do not have sufficient expertise to evaluate.

Working on Cell division in Plasmodium.

****Referee Cross-commenting****

I agree with the reviewers and some of the experiment suggested and the minor details have to be addressed. There are some loose ends and these suggestions will enhance clarity of the data. It is a very nice study and some of the comments suggested by reviewers will improve the manuscript.

Reviewer #3 (Evidence, reproducibility and clarity (Required)):

****Summary:****

The centrosome is the primary microtubule-organizing center (MTOC) in eukaryotic cells that nucleate spindle microtubules necessary for chromosome segregation. In most eukaryotic cells, the canonical centrosome is composed of centrioles surrounded by an electron-dense proteinaceous matrix named the pericentriolar matrix (PCM) competent for microtubule nucleation mitotic spindle assembly. Following the breakdown of the nuclear envelope breakdowns, the mitotic spindle microtubules gain access to the kinetochores of the condensed mitotic chromosomes. Once the mitotic spindle is fully developed, centrosomes are at opposite poles of the cells, and chromosomes are pulled toward opposite poles. Cell division completes with cytokinesis resulting in the active formation of two nuclei within two daughter cells. Interestingly, during its asexual replication cycle, the malaria parasite *Plasmodium falciparum* undergoes multiple asynchronous rounds of mitosis with segregation of uncondensed chromosomes followed by nuclear division within an intact nuclear envelope. The multi-nucleated cell is then subjected to a single round of cytokinesis that produces dozen of daughter cells. We know about the *Plasmodium* centrosome is that it is made of an acentriolar structure embedded in the nuclear envelope and serves as MTOC during cell division. However, the biogenesis and regulation of the *Plasmodium* centrosome are poorly understood. Given the peculiarity of the cell division in *Plasmodium* parasites, understanding the molecular mechanisms that drive and regulate MTOC duplication and maturation could unveil novel targets for the treatment of malaria. In this study, Simon et al. successfully applied challenging and cutting-edge microscopy techniques to monitor the dynamic formation of the spindle microtubules and MTOC during *Plasmodium* intraerythrocytic mitosis. In addition, they remarkably combined stimulated emission depletion (STED) with ultrastructure expansion microscopy to define an uncharacterized intranuclear compartment devoid of chromatin as the nucleation site of nuclear microtubules. And lastly, the authors adapted an in-resin correlative light and electron microscopy (CLEM) approach to define the centriolar plaque position in a novel intranuclear compartment with centrosomal function.

****Major comments:****

1. In the methods section, it is stated that across this study, three different anti-tubulin antibodies (alpha-tubulin B-5-1-2, alpha-tubulin TAT1, beta-tubulin KMX1) were used, and two anti-centrin antibodies (TgCentrin1 and PfCentrin3) were used, one of which seems to have been generated in this study (anti-PfCentrin3). It is unclear in the figures or results section when each of these antibodies was used, and the authors should give a rationale for using multiple antibodies in combination.

To label microtubules we used the mouse anti-alpha-tubulin B-5-1-2 (Sigma, T5168) antibody throughout the study. Except for U-ExM where we added two additional primary antibodies against tubulin. Due to the expansion of the samples the antibody binding epitopes are stretched out in space. This causes a significant reduction of local epitope concentration (expansion factor 4.5 in all directions results in ~ 80-fold increase in the volume), which can reduce the signal intensity. Adding multiple antibodies binding different epitopes of tubulin can compensate for this dilution effect to some degree, as has been shown before by Gao et al. 2018. At the same time the expansion contributes to the accessibility of the usually densely packed tubulin epitopes within the microtubule polymer, which certainly adds to the success of U-ExM. What the respective contributions of those effects are is not clear, but we found superior signal-to-noise ratios when combining three tubulin antibodies instead of using one. The TgCentrin1 antibody was only used in Fig. 2C (now Fig. 3B) and validated the localization pattern of our new PfCentrin3 antibody we used in the other pictures. We now provide clearer description of antibody usage in the methods section and a new supplemental table.

2. The anti-PfCentrin3 antibody seems to have been generated for this study. If this is the case, the authors should provide evidence that this antibody binds to the recombinant PfCentrin3 it was raised against and binds PfCentrin3 in parasite lysates.

The anti-PfCentrin3 antibody was, indeed, produced for this study and we should have provided our western blot data right away. We now show the requested blot, which shows bands at the appropriate size in parasite lysate as well as for the recombinant protein, in the supplements (Fig. S2, line 178).

3. In the first paragraph of the Results section, the authors' remark of centrin foci that they are "...only detectable later (Mov. S2) or sometimes not at all." In Figure 1 A-C, it is implied that the first observed division is the first nuclear division of that parasite. Given that some nuclei do not have a visible centrin focus, it cannot be concluded with certainty that these parasites only contain a single nucleus and that this is their first division. The authors would need to include a quantifiable DNA stain to show this unequivocally to show a single nucleus. It has undergone DNA replication, similar to Klaus et al., 2021 BioRxiv paper. In the absence of a DNA stain, the authors should reword to clarify that this is the first observed division and speculate that it is the first division of that nucleus, but the authors should draw no firm conclusions about the first division.

Indeed the variability in protein levels that can result from exogenous expression can lead to some cells not showing clear Centrin1-GFP foci. Although this is a rare event we wanted to acknowledge this observation. The live cell microtubule staining using Spy555-Tubulin we use is, however, highly specific and sensitive and would stain any nucleus undergoing division including the first one. If there would be more than one nucleus in the observed cell it would unequivocally show two clearly separated tubulin signals (hemispindle or mitotic spindle). To illustrate this we added Fig. 1B (line 148) showing two live parasites stained with SPY555-Tubulin plus a Hoechst-based dye showing one or two nuclei alongside the corresponding tubulin signal. We modified the text to clarify how we stage the parasite for time-lapse acquisition (line 154). We already extensively experimented with state of the art fluorogenic live cell DNA dyes (e.g. from Spirochrome and the Johnsson group) to visualize the nuclei directly in time lapse microscopy, but even at minimal concentrations they all significantly inhibit mitotic progression. We also add this information in the main text (line 150).

4. In the first paragraph of the Results section, the authors write: " We quantified the duration of hemispindle, accumulation and anaphase stages" Anaphase spindle fibers means that the sister chromatids are separated. In the absence of a centromeric marker like NDC80, it doesn't seem easy to claim the anaphase stage. The authors should write " extended spindle." The authors might also consider using the term collapsed spindle instead of accumulation to reflect the dynamic of the

intranuclear microtubules during the blood-stage replication. The same modification should be made for Figure 1B, so we read "hemispindle, collapsed spindle and extended spindle."

We thank the reviewer for this suggestion, which is very much in line with a comment by Reviewer 1 on the definition of anaphase. We acknowledge that the term is ill-defined here. Further, it suggests a mitotic morphology analogous to the one observed in "classical" models (prophase, metaphase, anaphase,...), which is not fully appropriate. Consequently, we decided to adapt the suggested terminology in Fig. 1 (and also new Fig. 2) and in the text (line 160).

5. Based on the evidence in this study, it cannot be stated unequivocally that the centrosome is entirely extranuclear, at least not as it is implied in Figure 3C. In Supplementary Figure 4, the microtubules appear to be extruding from a circular structure that may either be intranuclear or span the nuclear envelope. In Supplementary Figure 6, the structure pointed to as the centrosome appears to be embedded within the nuclear membrane with a top structure on the cytosolic side of the nuclear envelope. Thus, the best support for an extranuclear centrosome comes from the CLEM images. Still, it is noteworthy that the double membrane of the nuclear envelope is not visible on this slice in the region where the centrin fluorescence is found. Considering some of the fluorescence pixels for centrin are outside the parasite plasma membrane, and some of the Hoechst pixels are outside the nuclear envelope, this data does not show unequivocally that centrosomes are entirely extranuclear. However, this argument would be strengthened if the authors performed a proteinase K protection assay (or something similar) to determine if Centrin1 and Centrin3 are exposed to the cytosol. However, in the absence of that or further evidence, the authors should dampen their claims about the centrosome being exclusively extranuclear, as represented in the schematic in figure 3C.

We thank the reviewer for this comment, which highlights an issue in our communication of our working model of the centriolar plaque. At no point we intended to claim that the centrosome is exclusively extranuclear. Rather, centrin, which are currently the only reliable marker proteins, localize to a subcompartment of the extranuclear region of the centriolar plaque. Additionally, the centrosome clearly contains an intranuclear region. The composition of this intranuclear compartment is elusive, except that it harbors microtubule nucleation sites. Indeed, our model in Fig. 3C (now Fig. 4C) is misleading and not well annotated. The newly added NHS-Ester staining fortifies this claim (Fig. 3F-H). Consequently, we corrected our working model by adding an explicit figure labelling (now Fig. 4C).

We apologize for the misleading labelling in Fig. S6 (now Fig. S7). The green arrow was intended to point out the electron dense region associated with the nuclear membrane, which has been seen in previous studies, and was not intended to represent the entire extended centriolar plaque. If anything, this smaller region might provide the link between the intra and extranuclear compartments that the reviewer also identified in Suppl. Fig. 4 (now Fig. 2D). We modified the annotation of the Fig. S7 and Fig. 4A-B accordingly, labelling it the "electron dense region". More importantly, we hope that our newly added data using NHS-Ester staining of protein dense regions (Fig. 3F-H) highlights the spread of the centrosome across the nucleo-/cytoplasmic boundary more clearly.

Considering whether centrin is actually extranuclear, we feel that the data shown in Fig. 2A (now 3A) is convincing. We have, however, added two panels of the relevant regions showing centrin localization relative to the nuclear pore and adjusted the contrast as we acknowledge the limited "visibility" within the unadjusted panels. The fact, that the centrin signal slightly overlaps with the nuclear envelope in CLEM images can be explained by the relatively poor resolution of the widefield microscope we had to use to image the sections. From the other super-resolution images in the manuscript, we know that the

perimeter of the better resolved centrin signal is significantly smaller. Otherwise one had to assume from the CLEM data that centrin is also in the cytosol of the red blood cell and that DNA is localized outside the nucleus. On a similar note the fluorescence image is, contrary to the tomography image, a single slice since the thickness of the sample section (about 200nm) is significantly below the z-resolution (about 500nm) of a fluorescence microscope.

6. Throughout the study, the level of biological replication is unclear. The authors rigorously include all the data points for each of their graphs and the total number of images/videos quantified. And what needs to be added, in either the figure legends or a methods section, is the number of biological replicates for each of these measures came from.

We have added the number of replicas in the figure legends.

****Minor comments:****

7. STED is present as an acronym in the abstract and should be spelled out in full and clarified that it is a super-resolution microscopy technique.

We opted to remove STED from the abstract (leaving it at super-resolution, which includes expansion microscopy) to avoid disrupting the "flow" of the abstract and now spell out the acronym at the first mention in the introduction (line 127).

8. The second paragraph of the Results section states that ring and early trophozoite stage parasites do not express tubulin or centrin. Still, only an early trophozoite is shown in Supplementary Figure 2. Therefore, the authors should either include a similar image of a ring-stage parasite or remove ring-stage parasites from that statement.

We have removed the ring stages from the statement.

9. The second paragraph of the Results section contains the sentence, "At which point tubulin is reorganized into the bipolar microtubule array, which then forms the mitotic spindle cannot be resolved here." The authors are implying that the point at which tubulin is reorganized into the microtubule array, which goes on to form the mitotic spindle, cannot be resolved here. This is not particularly clear, though, and this sentence could be reworded for clarity.

We reformulated the sentence to clarify the point we failed to make with the previous wording (line 188).

10. The second paragraph of the Results section contains some statements about the results without referencing the figures that these statements come from. The authors should clarify this to make clear which figures each statement refers to.

We added more references to the appropriate figure throughout the paragraph (lines 188, 219, 223).

11. In the third paragraph of the introduction section, the authors write, "Centriolar plaques seem partially embedded in the nuclear membrane, but their positioning relative to the nuclear pore-like

"fenestra" remains unclear." Unfortunately, the lack of reference did not allow me to understand if the authors state literature or comment on past published results.

We added the reference which was incorrectly positioned before the sentence instead of at the end (line 82).

12. the authors could add some references:

- Second section of the introduction: " the 8-28 nuclei are packaged into individual daughter cells, called merozoites (Rudlaff et al. 2019 PMID: 31097714)
- Third section of the introduction: " The centrosome of *P.falciparum* is called centriolar plaque" (Arnot et al. 2011, Sinden 1991a); " the nuclear pore-like "fenestra" remains unclear (Wall et al. 2018; Zeeshan et al. 2020).
- Fourth section of the introduction: " tubulin antibody staining are extensive structures measuring around 2-4um (Ref?)
- When the authors introduce subpellicular microtubules of segmented schizonts, a reference to a study that shows these structures should be included.
- A previous study that shows the distinct structure of microtubule minus ends should be cited when this structure is described.
- Third section of the results, the authors should cite Bertiaux et al. 2021 with the Gambarotto et al. 2019 paper regarding U-ExM.

We apologize for missing some important references or putting them in the wrong position. We now added all the references or cite them again at the appropriate locations throughout the text.

13. Figure 1E shows hemispindle and mitotic spindle lengths of U-ExM expanded parasites, but the position within the figure and figure legend implies that these lengths were determined unexpanded parasites. Therefore, it should be stated in the figure legend that these measurements come from U-ExM expanded parasites. Moreover, I encourage the authors to include U-ExM images in the main figures. The images are beautiful, represent a significant technical achievement, and directly relate to Figure 1E. To the best of my knowledge, this is only the second study to perform expansion microscopy on *Plasmodium* and the first to use PFA-fixed parasites and a nuclear stain. It would be valuable for the *Plasmodium* and ExM communities to see this technical advancement represented in the main text.

We thank the reviewer for the appreciation of our ExM data and added it to Fig. 2B before the quantification of the microtubule length and number and added the information to the legend.

14. In the second paragraph of the Results section, the authors write, " but clearly display the microtubule cytoskeleton associated with the inner membrane complex." It would bring clarity to define in few words what the IMC is.

We included a short definition of the IMC (line 223).

15. The methods section details that the length of microtubules was determined by dividing the observed values by an expansion factor of 4.5. If the authors recorded the expansion factors of their gels, this data should be included, and how it was recorded should be stated in the methods. If not, the

authors should include the rationale of using an expansion factor of 4.5 as this is slightly different from the previously published expansion factor of *P. falciparum* of 4.3.

We recorded the expansion factor by measuring the gel size pre and post expansion with a ruler and found a factor of 4.5 on average. We added this information in the methods (line 688).

16. There are several parasite lines used in this study, and some figures are not clear what parasite line was used. Could the authors please include the parasite lines in the figure legends of Figure 1 D-F, Figure 3, Supplementary Figures 1-2, and Supplementary Figures 4-7?

We added the parasite line information in the legends as requested.

17. Nuclear pore complexes, of which Nup313 is a component, can have cytoplasmic, integral, and nuclear-facing components. If it has been shown previously that PfNup313 is the homolog of Nup214 in vertebrates present on the cytosolic side of NPC, this should be stated. If not, then it should be clarified that it is unknown whether Nup313 faces the cytoplasm, nucleus, or is embedded in the NE, as this has implications for the colocalization of Nup313 and Centrin.

Nuclear pore proteins are very poorly conserved in *P. falciparum* and Nup313 has only been recently identified as such (Kehrer et al. 2018) mainly by the presence of FG-repeats (as for all the other newly defined proteins). The only related ortholog that can be found through BLAST search against humans, yeasts, and *Arabidopsis* is Nup100 from *S. cerevisiae* (see alignment below). ScNup100 is a central pore localizing protein but the sequence similarity to Nup313 is low. We are not aware of any findings showing relatedness to vertebrate Nup214, while sequence analysis rather indicates the absence of orthology. To clearly demonstrate the individual positioning of the few known Nups within the parasite's nuclear pore complex would require a dedicated long-term project. However, due to the presence of FG-repeats one can assume that it is part of the central FG-Nups layer rather than of the intranuclear basket or the cytoplasmic filaments (line 255). Therefore it would localize more closely to the nuclear envelope than the latter. Either way, a clear gap between centrin and Nup313 signal can be identified and colocalization has not been observed. These data indicate that the exact position of Nup313 on the cytoplasmic, integral or nuclear-facing site is not decisive for the conclusions made in this study and our observations preclude scenarios where centrin is not extranuclear.

18. It seems from the image in Figure 2C that DRAQ5 and Hoechst have at least visually indistinguishable localizations. Have the authors taken any STED deconvolved images of nuclei stained with both Hoechst and DRAQ5? Considering the striking increase in detail of the Hoechst signal in STED deconvolved images, it may be informative both to this study and to people who work on chromatin organization what the chromatin staining looks like in the absence of bias towards chromatin state.

It would, indeed, be interesting to analyse chromatin organization by those means, but DRAQ5 is not a STED compatible dye, highly prone to bleaching, and therefore not suitable for such analysis. Being an infrared dye DRAQ5 is compared to the UV excited Hoechst also yielding a reduced spatial resolution, which is limited by the emitted wave length.

19. For the tomography and TEM images, the centrosome is indicated with an arrow, but it isn't entirely clear what that arrow is pointing to for some images. It would be clearer if the centrosome were outlined in green, like the NE, rather than just an arrow. This is particularly important for Supplementary Figure 4, where to my eye, it appears that the microtubules inside the chromatin-free region are coming directly out of a circular structure, which could be interpreted as the centriolar plaque.

The reviewer is right to point out the use of arrows for centrosome annotation. It was intended for orientation of readers to indicate the “likely position of the centriolar plaque” since a clear boundary around the centrosome can't be defined. It would have been more precise to indicate that the arrow is pointing at the electron dense region associated with the nuclear membrane, which is of course only one of the sub-regions of the centrosome. This is particularly important since we want to emphasize the extended dimensions of the centrosome. Consequently, we modified the annotation to “electron dense region” in all concerned figures and corresponding legends.

20. The ordering of Figure 2A-C seems to imply that the DNA-free region was measured in the STED deconvolved images, but the methods imply that it was in the confocal images. The authors should clarify this in the figure legend or by rearranging B and C's order.

Hoechst signal was indeed acquired and measured in confocal mode and to avoid confusion we have changed the order of the figures (now Fig. 3B-C) as suggested.

21. The authors should provide some more detail on how the DNA-free zone was measured. For example, was it measured on single slices or maximum intensity projections? Was it measured from the middle, far, or near side of the centrin focus? Etc.

The measurement was carried out in the slice where the DNA-free zone was in focus. Depth was measured from below the centrin signal until the “bottom” of the DNA-free zone. We hoped that the little schematic above the figure would clarify this question, but acknowledge the need to more clearly explain the measurement method, which we now do in the corresponding figure legend (Fig. 3C).

22. The methods state that the mCherry signal in figure 2C was detected using a mCherry nanobody. This should be clarified in the figure legends as it currently seems as if we see endogenous mCherry fluorescence.

The visible signal is certainly a combination of the mCherry plus the “boosting” effect from the Atto594-coupled nanobody that we added. Clearly, this should be mentioned in the figure legend, which we now do.

23. The data in Supplementary Figure 4 seems vital to the interpretation of the study. Therefore, for clarity, I encourage the authors to include Supplementary Figure 4 in Figure 2.

We share the reviewers view on these data and moved them to the main figures (now Fig. 3D).

24. In the last sentence of the discussion, it is unclear what the authors mean by how the nuclear compartment "splits," could they please clarify?

We were referring to the event of centrosome duplication, which has to occur during nuclear division. In a structure without centrioles or a spindle pole body structure forming a half bridge we therefore need a new model to explain how the two poles of the spindle are formed. Potential modes are splitting or de novo assembly. This aspect, as also pointed out by other reviewer, warrants a bit more explanation, which can now be found in the discussion (line 468).

25. If the pArl-PfCentrin3-GFP plasmid or pDC2-cam-coCas9-U6.2-hDHFR have been published previously, the respective studies should be cited. If not, the study where the vector backbones were first established should be cited.

We have now cited the original studies publishing the vector backbones for the first time in the methods (lines 490, 501).

26. From the current text, it is not clear that the Nup313 tagged parasites also had a GImS ribozyme. It is shown in Supplementary Figure 3, but the authors should clarify either in the text of the results, or figure legends, that this parasite line was Nup313_3xHA_GImS

The Nup313-tagged line indeed has a glms ribozyme after the HA-tag, which we now mention in the figure legends.

27. In the plasmid constructs section of the methods, the authors list several primers by number but not by sequence. Instead, the authors should include the sequence and orientation of each of the primers mentioned in a table as supplementary data.

This is a good suggestion. We have generated a table at the end of the supplementary data file and on this occasion we also added tables of all the antibodies and dyes used in this study.

28. The authors should cite the study where the TgCentrin1 antibody was generated and provide the Rat anti-HA 3F10 antibody catalog number, as catalog numbers are provided for other commercial primary antibodies.

We now provide the missing catalog numbers in the supplemental data table.

There is an issue with the formatting of the journal-title in the Kukulski et al. reference.

Thank you for noticing this error, which we now corrected.

Reviewer #3 (Significance (Required)):

The genome of *P. falciparum* is fully sequenced; however, over 50% of encoded proteins are of unknown function, with many of these proteins unique to Plasmodium parasites. By identifying and characterizing essential biological processes, especially those divergent from human host cell processes, we will formulate ways to interfere with them by developing novel antimalarial drugs. The process of Plasmodium cell division differs from the classical cell cycle of its human host. In the study led by Caroline Simon, authors successfully utilized recent developments of super-resolution microscopies on expanded parasites to identify novel features of cell division machinery of the malaria blood-stage parasite.

Simon et al.'s work highlight the growing interest in the diversity of cell division mode of Apicomplexan parasites, which will likely contribute to a deeper understanding of the origin and functional role of the centrosome in eukaryotic life. In 2020, the Open Biology journal published a unique article collection named Focus on Centrosome Biology showcasing research that advanced our knowledge on centrosome function, evolution and abnormalities. In addition, the reported findings will interest research groups studying cell cycle regulation and evolution beyond the field of parasitology.

Our lab studies the peculiar cell cycle of *Plasmodium falciparum* to gain a functional understanding of mechanistic principles of nuclear envelope assembly and integrity during the cell division of the human malaria parasite.

****Referee Cross-commenting****

It is a wonderful study, and once all reviewer's comments are addressed, the manuscript should be in excellent shape for publication.

August 24, 2021

RE: Life Science Alliance Manuscript #LSA-2021-01199

Dr. Julien Guizetti
Heidelberg University Hospital
Unknown
Germany [DE]

Dear Dr. Guizetti,

Thank you for submitting your revised manuscript entitled "An extended DNA-free intranuclear compartment organizes centrosomal microtubules in *Plasmodium falciparum*". We would be happy to publish your paper in Life Science Alliance pending final revisions necessary to meet our formatting guidelines.

- please upload your supplementary figures as single files as well
- please add a Running Title to our system
- please add a Summary Blurb/Alternate Abstract in our system
- please add a Category for your manuscript in our system
- please add the Twitter handle of your host institute/organization as well as your own or/and one of the authors in our system
- please add the contribution of all Authors to our system
- please use the [10 author names, et al.] format in your references (i.e. limit the author names to the first 10)
- please add your supplementary figure and table legends to the main manuscript text after the main figure legends
- please upload your Tables in editable .doc or excel format
- please upload supplementary movies accordingly and add callouts to the manuscript text
- please add a callout for Supplementary table 2 to the main manuscript text
- please add Data Availability section and Approval Statement

LSA now encourages authors to provide a 30-60 second video where the study is briefly explained. We will use these videos on social media to promote the published paper and the presenting author. Corresponding or first-authors are welcome to submit the video. Please submit only one video per manuscript. The video can be emailed to contact@life-science-alliance.org

A. FINAL FILES:

B. MANUSCRIPT ORGANIZATION AND FORMATTING:

Sincerely,

September 3, 2021

RE: Life Science Alliance Manuscript #LSA-2021-01199R

Dr. Julien Guizetti
Centre for Infectious Diseases, Heidelberg University Hospital
Unknown
Germany [DE]

Dear Dr. Guizetti,

Thank you for submitting your Research Article entitled "An extended DNA-free intranuclear compartment organizes centrosome microtubules in malaria parasites". It is a pleasure to let you know that your manuscript is now accepted for publication in Life Science Alliance. Congratulations on this interesting work.

DISTRIBUTION OF MATERIALS:

Again, congratulations on a very nice paper. I hope you found the review process to be constructive and are pleased with how the manuscript was handled editorially. We look forward to future exciting submissions from your lab.

Sincerely,
